# Differentiating between integration and non-integration strategies in perceptual decision making

**Gabriel M Stine[1]\*, Ariel Zylberberg[2,3], Jochen Ditterich[4], Michael N Shadlen[1,2,5]**

[1]Department of Neuroscience, Columbia University, New York, United States; [2]Mortimer B. Zuckerman Mind Brain Behavior Institute and The Kavli Institute for Brain Science, Columbia University, New York, United States; [3]Department of Brain and Cognitive Sciences, University of Rochester, Rochester, United States; [4]Center for Neuroscience and Department of Neurobiology, Physiology & Behavior, University of California, Davis, United States; [5]Howard Hughes Medical Institute, Columbia University, New York, United States

**Abstract** Many tasks used to study decision-making encourage subjects to integrate evidence over time. Such tasks are useful to understand how the brain operates on multiple samples of information over prolonged timescales, but only if subjects actually integrate evidence to form their decisions. We explored the behavioral observations that corroborate evidence-integration in a number of task-designs. Several commonly accepted signs of integration were also predicted by non-integration strategies. Furthermore, an integration model could fit data generated by non-integration models. We identified the features of non-integration models that allowed them to mimic integration and used these insights to design a motion discrimination task that disentangled the models. In human subjects performing the task, we falsified a non-integration strategy in each and confirmed prolonged integration in all but one subject. The findings illustrate the difficulty of identifying a decision-maker's strategy and support solutions to achieve this goal.

**\*For correspondence:**
gabriel.stine@columbia.edu

**Competing interests:** The authors declare that no competing interests exist.

## Introduction

Unlike reflexive behaviors and simple sensorimotor response associations, cognitive functions are not beholden to fleeting sensory information or the real-time control of motor systems. They incorporate many samples of information, spanning timescales of tenths of seconds to years. A fruitful approach to study how the brain operates on information over prolonged timescales has been to record and perturb neural activity while subjects make decisions about noisy sensory stimuli (*Shadlen and Kiani, 2013*). The presence of this noise encourages subjects to reduce it by integrating the sensory information over time. If the timescale of integration is long enough, then the neural mechanisms of this decision process are likely to provide insight into those that allow cognitive processes to operate over long timescales.

One concern is that a subject may not integrate sensory evidence when making these decisions, even though they are encouraged to do so. The concern is exacerbated in animal subjects because they must learn the task without explicit instructions. Experimentalists are therefore posed with a challenge: they must infer a subject's decision strategy from behavioral measurements like choice-accuracy and reaction time (RT). Correct identification of a subject's strategy is especially important for neuroscience, which aims to elucidate biological mechanisms of evidence integration. Results cannot bear on the neural mechanisms of evidence integration if subjects were not integrating evidence.

Mathematical models of evidence integration provide a framework for using behavioral measurements to infer a subject's decision strategy. These models posit that samples of noisy evidence are integrated over time until a threshold is exceeded or until the stream of evidence (e.g. the stimulus) is extinguished, at which point a decision is made. Examples include drift-diffusion, race, and attractor models (*Gold and Shadlen, 2007*; *Deco et al., 2013*). Integration models predict several observations that are common in behavioral data: in fixed stimulus duration (FSD) tasks, subjects' decisions appear to incorporate evidence presented throughout the entire stimulus presentation epoch (e.g. *Odoemene et al., 2018*; *Deverett et al., 2018*; *Morcos and Harvey, 2016*; *Katz et al., 2016*; *Yates et al., 2017*; *Wyart et al., 2012*); in variable stimulus duration (VSD) tasks, accuracy improves with increasing stimulus duration (e.g. *Britten et al., 1992*; *Gold and Shadlen, 2000*; *Kiani et al., 2008*; *de Lafuente et al., 2015*; *Kiani and Shadlen, 2009*; *Bowman et al., 2012*; *Brunton et al., 2013*; *Robertson et al., 2012*); and, in free response (FR) tasks, RTs for the most difficult stimulus conditions are longer than those for easier stimulus conditions (e.g. *Roitman and Shadlen, 2002*; see *Ratcliff and McKoon, 2008* for review). Indeed, the fits of integration models to behavioral data are often remarkably precise. These observations are commonly adduced to conclude that a subject integrated evidence.

Yet, it is unclear whether these observations reveal a subject's actual decision strategy. For example, previous work has shown that models that posit little to no integration can also fit data from FR tasks (*Ditterich, 2006*; *Thura et al., 2012*). This raises a critical question: which behavioral observations corroborate an integration strategy? In cases where integration cannot be differentiated from strategies that lack integration, it will be important to identify why, as doing so may aid the design of experiments that encourage integration.

We compared the predictions of evidence integration to those of strategies that lack integration in a number of task-designs. We found that many signatures of evidence integration are also predicted by non-integration strategies, and we identified the critical features that allowed them to mimic integration. We used these insights to design a novel variant of the random-dot-motion discrimination task that disentangles evidence integration from non-integration strategies. With this task-design, we ruled-out a non-integration strategy in each subject and confirmed prolonged integration times in all but one. Our results underscore the difficulty of inferring subjects' strategies in perceptual decision-making tasks, offer an approach for doing so, and illustrate the importance of evaluating strategies at the level of individual subjects.

## Results

We explored the observations predicted by an evidence integration strategy and those predicted by two non-integration strategies in binary perceptual decision-making tasks. The main model representing an integration strategy was a variant of the drift-diffusion model (*Ratcliff and McKoon, 2008*) that has been used extensively to explain behavioral data from random-dot-motion discrimination tasks (*Figure 1A*). For simplicity, we refer to this model as *integration*. It posits that noisy evidence is sequentially sampled from a stationary distribution of random values representing the noisy momentary evidence from the stimulus and is perfectly integrated (i.e. with no leak). The decision can be terminated in two ways: either the integrated evidence exceeds one of two decision-bounds (the timing of which determines the decision's duration) or the stimulus is extinguished. In both cases, the choice is determined by the sign of the integrated evidence.

The first non-integration model we considered was *extrema detection* (*Figure 1B*). The model was inspired by probability summation over time (*Watson, 1979*), which was proposed as an explanation of yes-no decisions in a detection task. *Extrema detection* is also similar to other previously proposed models (*Cartwright and Festinger, 1943*; *Ditterich, 2006*; *Cisek et al., 2009*; *Brunton et al., 2013*; *Glickman and Usher, 2019*). In the extrema detection model, evidence is sampled sequentially from a stationary distribution until a sample exceeds one of two decision-bounds, which terminates the decision. Crucially, however, the sampled evidence is not integrated. Evidence that does not exceed a decision-bound has no effect on the decision; it is ignored and forgotten. If an extremum is detected, the process is terminated and no additional evidence is considered. If the evidence stream extinguishes before an extremum is detected, then the choice is made independent of the evidence (i.e. random guessing).

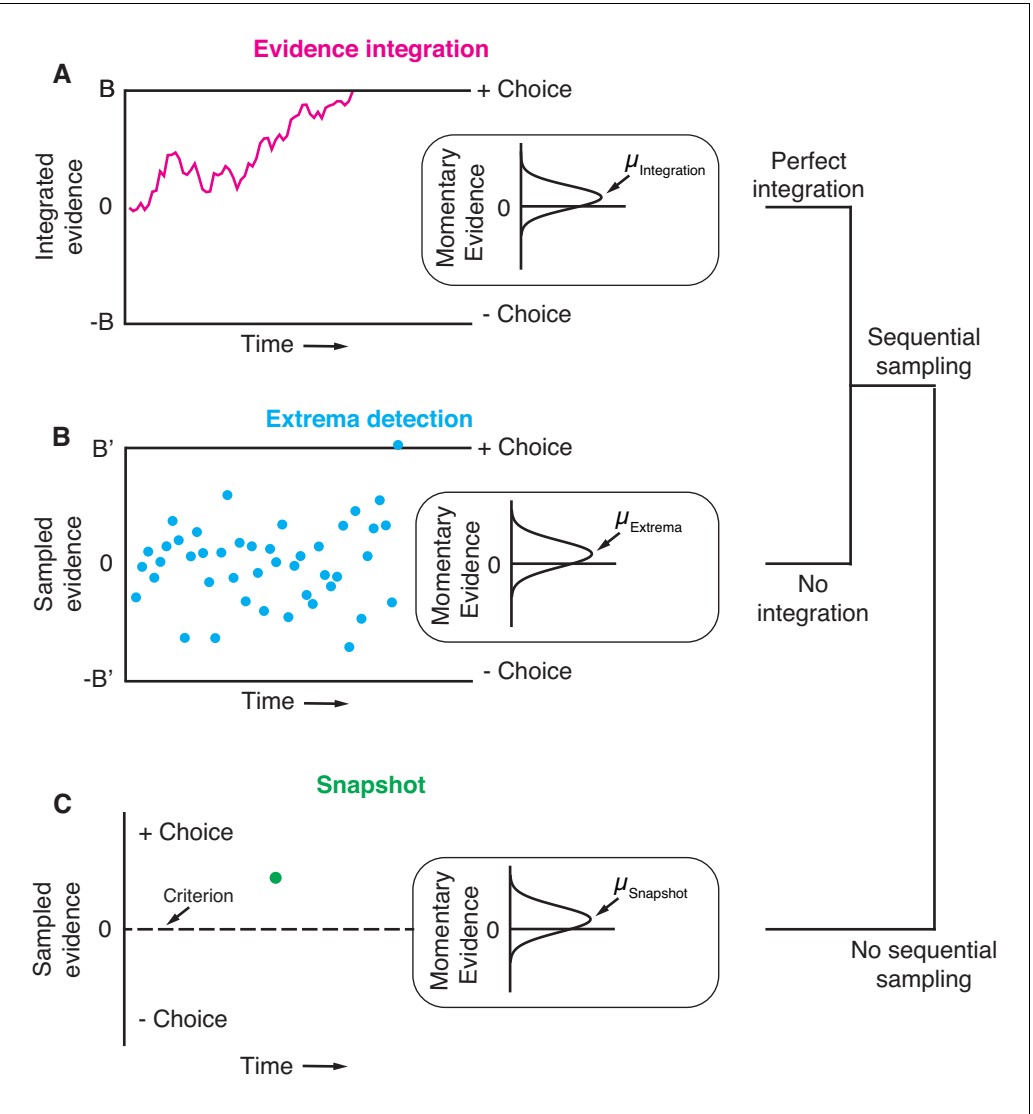

**Figure 1.** Three general decision-making models. Each schematic represents the evidence that resulted in a single, positive choice. (A) The evidence integration model. Sequential samples of noisy momentary evidence are integrated over time until a decision-bound is reached, resulting in a positive or negative choice. The momentary evidence is assumed to be sampled from a Gaussian distribution with mean proportional to the stimulus strength. If the stimulus is extinguished before a decision-bound is reached, then the decision is based on the sign of the integrated evidence. (B) The extrema detection model. Momentary evidence is sequentially sampled but not integrated. A decision is made when one of the samples exceeds a detection threshold (i.e. an extremum is detected). Samples that do not exceed a detection threshold are ignored. If the stimulus is extinguished before an extremum is detected then the choice is determined randomly. (C) The snapshot model. Momentary evidence is neither sequentially sampled nor integrated. Instead, a single sample of evidence (i.e. a snapshot) is acquired on each trial and compared to a decision criterion to render a choice. The sampling time is random and determined before the trial begins.

The second non-integration model, what we term *snapshot*, not only lacks integration but also sequential sampling. The decision-maker acquires a single sample of evidence at a random time during the stimulus presentation. This single sample of evidence is compared to a decision criterion in order to resolve the choice. The distribution of sampling times is not constrained by any mechanism and can thus be inferred to best match data. Similar to *extrema detection*, if the evidence stream extinguishes before a sample is acquired, then the choice is determined by a random guess. To facilitate comparison, we parameterized the three models as similarly as possible, such that they were

conceptually nested. In other words, *extrema detection* only differed from *integration* in its lack of integration, and *snapshot* only differed from *extrema detection* in its lack of sequential sampling. We assumed flat decision-bounds in the integration and extrema detection models, unless stated otherwise.

In the first part of this paper, we simulated each model in fixed stimulus duration, variable stimulus duration, and free response task-designs to identify observations that differentiate between integration and non-integration strategies. On each simulated trial, the models specified a positive or negative choice based on noisy sensory evidence. We also tested whether an integration model can fit simulated data generated by non-integration models. Our primary focus with this complementary approach was not to validate (or invalidate) that a model comparison metric favors the non-integration model, but to ask whether the integration fit could lead to erroneous conclusions had the non-integration models not been considered. In the second part of the paper, we used the insights gained from the first part to identify the decision strategies of subjects performing a random-dot-motion discrimination task.

## Integration and non-integration strategies are difficult to differentiate

### Fixed stimulus duration tasks

In FSD task-designs, the sensory stimulus is presented for a fixed, experimenter-controlled duration on every trial and subjects report their choice after viewing the stimulus. The stimulus strength typically varies across trials such that the correct choice ranges from obvious to ambiguous. Therefore, if a subject performs well and integrates evidence, they should exhibit a sigmoidal psychometric function that saturates at near-perfect accuracy when the stimulus is strong.

To infer integration, the experimenter also exploits the stochastic aspects of the stimulus and attempts to ascertain when, during the stimulus presentation, brief fluctuations in stimulus strength about the mean exerted leverage on the decision. The strength of evidence is represented as a time series, and the experimenter correlates the variability in this value, across trials, with the decision. This is achieved by averaging across the trials ending in either decision and subtracting the averages or by performing logistic regression at discrete time points spanning the stimulus duration. We refer to the outcome of either approach as a psychophysical kernel (*Figure 2B*). It is a function of time that putatively reflects a subject's temporal weighting profile, or the average weight a subject places on different time-points throughout the stimulus presentation epoch (cf. *Okazawa et al., 2018*). The shape of the psychophysical kernel is thought to be informative of decision strategy because a given strategy often predicts a specific temporal weighting profile. For example, a subject that perfectly integrates evidence weights every time-point in the trial equally, and so they ought to have a flat psychophysical kernel. In a FSD task-design, the observations of a flat psychophysical kernel and successful task-performance (i.e. a sigmoidal psychometric curve) are commonly cited as evidence for an integration strategy (e.g. *Shadlen and Newsome, 1996*).

We found that these observations also arise from non-integration models. We simulated *integration*, *extrema detection*, and *snapshot* (with a uniform sampling distribution) in a FSD task-design and asked whether *extrema detection* and *snapshot* can generate data that mimics data generated by *integration*. As shown in *Figure 2A*, the extrema detection and snapshot models can produce sigmoidal psychometric curves whose slope matched that of the integration model.

Given the simulated choice-data, all three models produced psychophysical kernels that are effectively indistinguishable (*Figure 2B*). To calculate a psychophysical kernel for each model, the simulations included a small, 100 ms stimulus pulse whose sign and timing were random on each trial (see Materials and methods for details). The kernel was calculated by determining the pulse's effect on choices as a function of time, as defined by a logistic regression (*Figure 2B*). The non-integration models posit that only a very short time-period during the stimulus epoch contributes to the choice on each trial. Yet, their psychophysical kernels misleadingly suggest that evidence presented throughout the entire stimulus epoch contributed to choices. These results held for a range of generating parameters (*Figure 2—figure supplement 1*). We thus conclude that the observations of high choice-accuracy and a flat psychophysical kernel are not, on their own, evidence for or against any particular decision-making strategy.

Why are *extrema detection* and *snapshot* able to mimic *integration* in a FSD task? First, they can match the choice-accuracy of the integration model because of the lack of constraints on how the

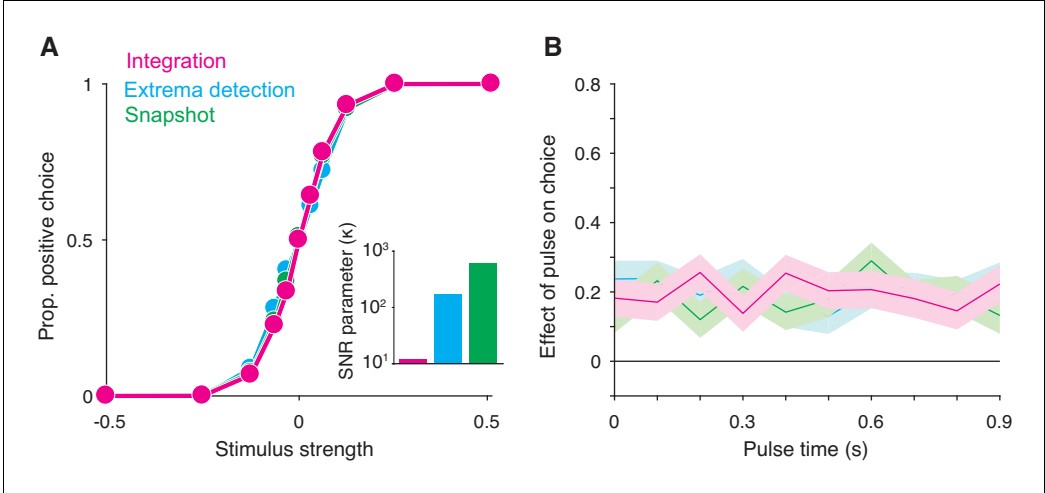

**Figure 2.** Integration and non-integration models produce similar psychometric functions and psychophysical kernels in a fixed stimulus duration task (stimulus duration = 1 second). (**A**) Proportion of positive choices as a function of stimulus strength for each model simulation (N = 30,000 trials per simulation). The inset displays the SNR parameter ($\kappa$) for each of the three models. All three models are capable of producing sigmoidal psychometric functions of similar slope but require different SNR parameters. (**B**) Psychophysical kernels produced from the choice-data in **A**. Each simulated trial contained a small, 100 ms long stimulus pulse that occurred at a pseudorandom time during the trial. Kernels are calculated by computing the pulse's effect on choices (coefficients of a logistic regression, *Equation 16*) as a function of pulse time. Shaded region represents the standard error. Like integration, non-integration models are capable of producing equal temporal weighting throughout the stimulus presentation epoch.

The online version of this article includes the following figure supplement(s) for figure 2:

**Figure supplement 1.** Extension of the exercise in *Figure 2* to a low sensitivity regime (**A**) and a high sensitivity regime (**B**).

sensory stimulus is transformed into a signal-to-noise ratio (SNR) of the momentary evidence. In each model, this transformation is determined by a single parameter, $\kappa$. In many cases, the SNR cannot be measured directly and thus the true SNR is generally unknown. Each model's $\kappa$ parameter is therefore free to take on any value. This bestows *extrema detection* and *snapshot* with the ability to produce choice-accuracy that matches that of *integration*; while these models are highly suboptimal compared to the integration model, they can compensate by adopting higher SNR (Figure 2A, inset).

Nevertheless, this trade-off does not explain why *extrema detection* and *snapshot* can produce flat psychophysical kernels. *Snapshot* can produce a flat kernel—and theoretically any other kernel shape—because the data analyst is free to assert any distribution of sampling times. The shape of the desired psychophysical kernel can thus be used to infer the shape of the distribution of sampling times. To generate a flat kernel, we used a uniform distribution for the sampling times.

It is less intuitive why *extrema detection* can predict a flat kernel. Indeed, in *extrema detection*, the sample of evidence that determines the choice is exponentially distributed in time (see Materials and methods). This implies that the model should produce early temporal weighting that decays toward zero. The degree of early weighting is governed by the $\kappa$ and decision-bound parameters, the combination of which determines the probability of detecting an extremum on each sample for a given stimulus strength. If this probability is high, then it is very likely that an extremum will be detected early in the trial. In contrast, if this probability is low enough, then detecting an extremum late in the trial will only be slightly less likely than detecting it early in the trial. A low detection probability also leads to more trials that end before an extremum is detected, in which case the choice is determined by random guessing. These guess trials effectively add noise centered at zero weighting to the kernel. With this noise, it is exceedingly difficult to distinguish the very slight, early weighting from flat weighting, even with tens of thousands of trials.

## Variable stimulus duration tasks

A major benefit of integrating information over time is that a decision-maker can reduce noise through averaging. This leads to a prediction: a subject's sensitivity (i.e. accuracy) should improve if they are given more time to view the stimulus, and thus average-out more noise. More precisely, if a subject is perfectly integrating independent samples of evidence, then the improvement in sensitivity should be governed by the square-root of the stimulus duration. This prediction can be tested with a VSD task, in which the experimenter-controlled stimulus duration on each trial varies randomly. Indeed, sensitivity often improves with increasing stimulus duration and often does so at the rate predicted by perfect integration (*Kiani et al., 2008*; *Brunton et al., 2013*). These observations are complemented by the fact that integration models fit VSD data well. If these observations can be relied on to conclude that a subject was integrating evidence, then they should be absent for data generated by non-integration models.

Yet, simulations reveal that these observations are also predicted by non-integration models. We simulated VSD data with either an extrema detection model or a snapshot model (with an exponential sampling distribution). The figure shows that the sensitivity of both models improved with increasing stimulus duration, although sensitivity plateaued for the longer stimulus durations (data points in *Figure 3A and B*, respectively).

Unlike *integration*, the extrema detection and snapshot models do not improve their sensitivity by averaging-out noise. Instead, the improvement of sensitivity is attributed to the guessing mechanisms posited by the models. If the stimulus extinguishes before an extremum is detected or a sample is acquired, then the models' decisions are based on a coin flip. Thus, the simulated data result from two types of trials: ones in which decisions are based on sampled evidence, and ones in which decisions result from random guessing. The models predict that sensitivity should improve with time because guesses are less likely with longer stimulus durations (*Figure 3C–D*).

Given this integration-like behavior of *extrema detection* and *snapshot*, we wondered whether an integration model could successfully—and hence misleadingly—fit the simulated datasets. *Figure 3A and B* show fits of the integration model (magenta curves) to the simulated extrema detection and snapshot data, respectively. Note that the model fits deviate from perfect integration at the longest stimulus durations because of the decision-bound parameter, which allows a decision to be made before the stimulus extinguishes (see *Kiani et al., 2008*). Qualitatively, *integration* provided an excellent account of the simulated extrema detection dataset. Indeed, the agreement between the integration model and the data might lead an experimenter to erroneously conclude that the data were generated by an integration strategy. The fit of the integration model to the simulated snapshot dataset was noticeably worse (*Figure 3B*). We also compared the fit of the integration model to that of the corresponding data-generating model (*Figure 3A–B*, dashed curves) using a standard model comparison metric. For both datasets, a model comparison unsurprisingly favored the data-generating model over the integration model ($\Delta\mathrm{BIC} = -70.34$ when *extrema detection* generated the data; $\Delta\mathrm{BIC} = -182.79$ when *snapshot* generated the data). We found similar results when *integration* served as the data-generating model (*Figure 3—figure supplement 1A*); the extrema detection model fit the simulated data well, whereas the snapshot model did so considerably worse.

In our implementations of *extrema detection* and *snapshot*, choices are determined by random guessing if the stimulus extinguishes before an extremum is detected or a sample is acquired. We considered whether a different rule for choices in this condition would lead to different conclusions. One alternative is a 'last-sample' rule, in which the sign of the final evidence sample determines the choice. Similar rules are often implicit in models that implement a non-integration strategy with high levels of leak (e.g. the 'burst detector' model from *Brunton et al., 2013*). For *snapshot*, a last-sample rule eliminated the model's ability to improve its sensitivity with time because expected performance is independent of when the snapshot is acquired. However, this was not the case for the extrema detection model. *Figure 3—figure supplement 1B* shows the fit of the last-sample model to the simulated data in *Figure 3A*, which was generated by an extrema detection model that used the guess rule. The last-sample rule still predicts sensitivity that increases with stimulus duration, but with a shallower slope. Interestingly, the fit was worse than that of the integration model ($\Delta\mathrm{BIC} = 8.73$), even though the last-sample model is conceptually similar to the data-generating model. As is the case in a FSD task-design, the guessing mechanism is essential to *extrema detection's* mimicry of *integration* in a VSD task-design.

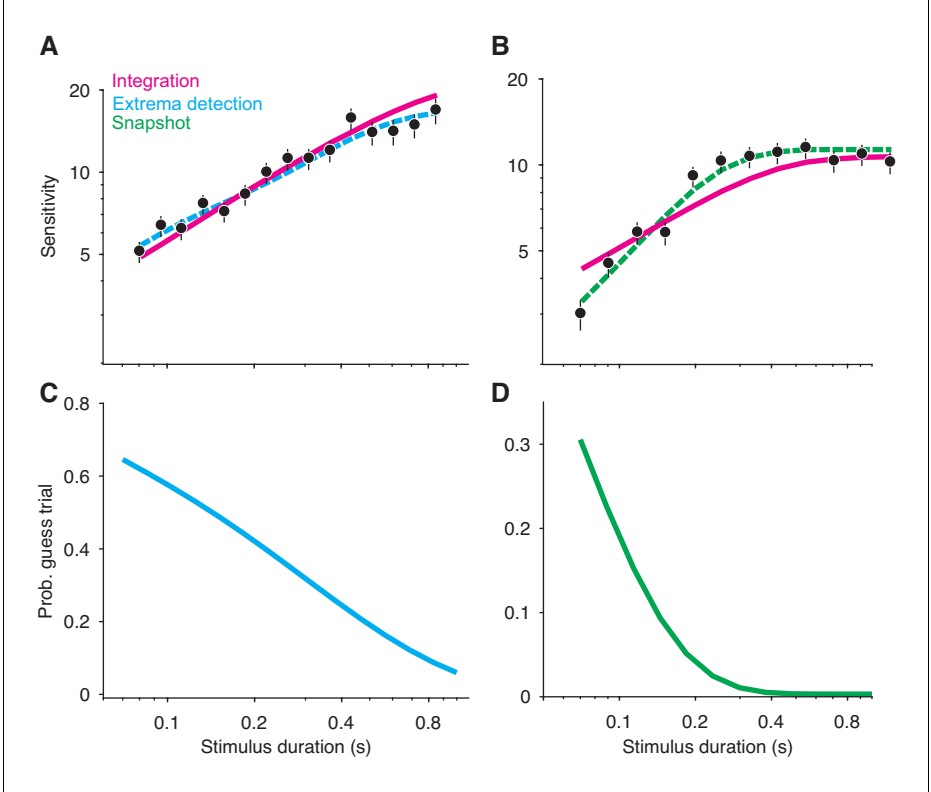

**Figure 3.** Non-integration models mimic integration in a variable stimulus duration (VSD) task. (**A**) Sensitivity as a function of stimulus duration for simulated data generated from an extrema detection model (black data points; N = 20,000 total trials). Sensitivity is defined as the slope of a logistic function (*Equation 18*) fit to data from each stimulus duration (error bars are s.e.). The dashed cyan line represents the data-generating function (extrema detection). The solid magenta line represents the fit of the integration model to the simulated data. Although the data were generated by an extrema detection model, there is a close correspondence between the data and the integration fit. (**B**) The same as in (**A**), except the data were generated by a snapshot model. (**C,D**) The probability of a guess trial as a function of stimulus duration. Extrema detection (**C**) and snapshot (**D**) mimic integration in a VSD task in part because of the 'guess' rule. If the stimulus is extinguished before an extremum is detected or a snapshot is acquired, the choice is determined by a random guess.

The online version of this article includes the following figure supplement(s) for figure 3:

**Figure supplement 1.** Integration as the data-generating model (A) and exploration of a last-sample rule in extrema detection (B).

## Free response tasks

In a FR task, subjects are free to report their decision as soon as they are ready, thus furnishing two measurements on each trial: the subject's choice and reaction time (RT; relative to stimulus onset). Note that any model designed to explain data from a FR task must prescribe how a decision is terminated (e.g. a decision bound). Additionally, models generally posit that the measured RT is the sum of the duration of two processes: (*i*) the decision process—the evaluation of evidence up to termination—and (*ii*) a set of operations, unrelated to the decision, comprising sensory and motor delays. The durations *i* and *ii* are termed the decision time and the non-decision time, respectively. Bounded evidence integration explains why decisions based on weaker evidence are less accurate and take longer to make, and it can often explain the precise, quantitative relationship between a decision's speed and accuracy (*Gold and Shadlen, 2007*; *Ratcliff and McKoon, 2008*).

Could FR data generated from a non-integration model be mistakenly attributed to *integration*? Several analyses and model fitting exercises on simulated data demonstrate that this is indeed possible. First, *extrema detection* also predicts that RT should depend on the stimulus strength (*Figure 4A*, top; see also *Ditterich, 2006*): the weaker the stimulus strength, the more samples it

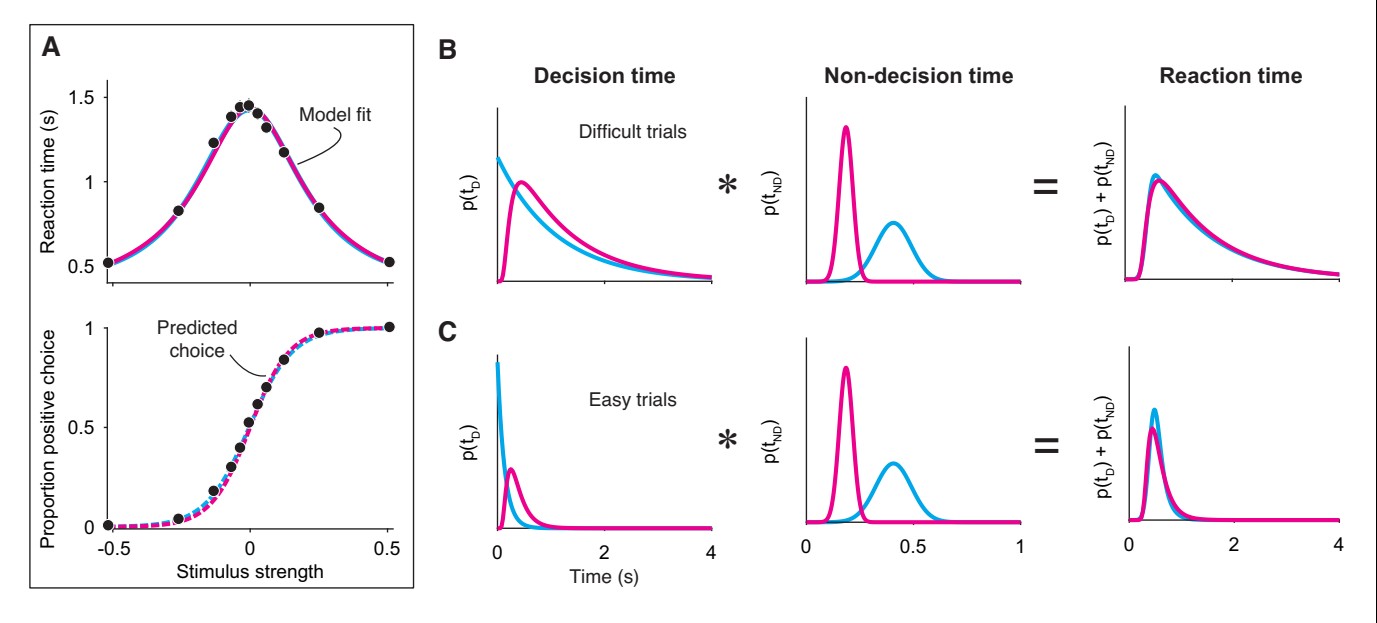

**Figure 4.** Integration and extrema detection behave similarly in free response tasks. (**A**) Simulated choice-reaction time (RT) data generated by the extrema detection model (black data points; N = 19,800 total trials, 1800 trials per stimulus strength). RT (top) and the proportion of positive choices (bottom) are plotted as a function of stimulus strength. Solid colored curves are fits of the integration (magenta) and extrema detection (cyan) models to the mean RTs. Dashed curves are predictions using the parameters obtained from the RT fits. Note that the models' predictions are indistinguishable. (**B,C**) The models produce similar RT distributions (right) but predict different decision times (left) and non-decision times (middle). The predicted RT distribution is the convolution (denoted by *) of the decision time distribution with the non-decision time distribution. (**B**) Depicts these distributions for difficult trials (stimulus strength = 0) and (**C**) depicts these for easy trials (stimulus strength = ±.512).
The online version of this article includes the following figure supplement(s) for figure 4:

**Figure supplement 1.** *Extrema detection* can fit RT means and predict choice-accuracy when *integration* serves as the data-generating model.
**Figure supplement 2.** Model comparison between *integration* and *extrema detection* for simulated data.

takes for an extremum to be detected. In contrast, the snapshot model does not predict this—and hence does not mimic *integration*—because the time at which a sample is acquired is independent of stimulus strength. For this reason, we do not include *snapshot* in subsequent analyses.

Second, it is possible for *integration* to successfully predict choice-accuracy from mean RTs, even though the data were generated by an extrema detection model. The integration model with flat bounds has separate, closed-form equations that describe the RT and choice functions given a set of model parameters (see Materials and methods). This allows us to estimate the model parameters that best fit the mean RT data, which can then be plugged into the equation for the choice function to generate predictions for the choice-data (*Shadlen and Kiani, 2013*; *Kang et al., 2017*; *Shushruth et al., 2018*). Conveniently, the same procedure can be performed using the equations derived for the extrema detection model (also with flat bounds; see Materials and methods). The top panel in *Figure 4A* shows the fits of both models to the simulated mean RTs (solid curves), and the bottom panel displays the resulting choice-accuracy predictions (dashed curves). The predictions of both models are remarkably accurate, and, the models are indistinguishable on the basis of either the RT fits or the choice-accuracy predictions. Further, the similarity of the models' behavior does not depend on which is the data-generating model; *extrema detection* can fit RT means and predict choice-accuracy when *integration* serves as the data-generating model (*Figure 4—figure supplement 1*). Thus, although an integration model might accurately fit—and even predict—data from a FR task-design, strong conclusions may not be warranted. Later, however, we will show that choice-accuracy predictions can be informative with just one additional constraint.

Finally, *integration* and *extrema detection* are not distinguishable on the basis of the shapes of the RT distributions. Because the models assume that RTs are determined by the sum of the decision time and the non-decision time, the predicted distribution for RT is the convolution of the decision

time distribution with the non-decision time distribution (here, assumed to be Gaussian). As described earlier, the extrema detection model posits decision times that are exponentially distributed. Thus, one might be tempted to rule-out this model because RTs do not typically conform to an exponential distribution. However, after convolving the exponential decision time distribution with the Gaussian non-decision time distribution, the resulting RT distribution is similar to that of the integration prediction and to what is typically observed in data (*Figure 4B and C*). Indeed, the ex-Gaussian distribution—precisely what the extrema detection model predicts—is often used as a descriptive model of RT distributions (*Luce, 1986*; *Ratcliff, 1993*; *Whelan, 2008*). We fit the integration and extrema detection models to the RT distributions simulated by *extrema detection*. Both models fit these RT distributions reasonably well (by eye), although a model comparison favored *extrema detection* ($\Delta$BIC $= -316.41$).

A more systematic model-comparison further illustrates that the models are difficult to disentangle in a FR task, especially when there is a limited number of trials. We simulated 600 datasets, half of which were generated by *integration* and the other half of which were generated by *extrema detection*. Each dataset comprised 10, 100, or 1000 trials per signed stimulus strength and the generating parameters were chosen pseudorandomly within a range of plausible parameter values (see Materials and methods). We then fit the full RT distributions of each simulated dataset with the integration and extrema detection models and calculated a $\Delta$BIC statistic for each comparison (*Figure 4—figure supplement 2*). With 10 trials per stimulus strength (120 trials in total), the large majority of $\Delta$BICs did not offer strong support for either model ($\Delta$BIC<10 in 181 of 200 datasets) and a large proportion of datasets with 100 trials per stimulus strength (1200 trials in total) still did not yield strong support for either model ($\Delta$BIC<10 in 63 out of 200 datasets). In contrast, all but one dataset with 1000 trials per stimulus strength (12,000 trials in total) yielded strong support for the data-generating model.

While *extrema detection* and *integration* predict similar RT distributions, *extrema detection* predicts shorter decision times and longer non-decision times. Because decision times are exponentially distributed in the extrema detection model, they are skewed toward shorter times compared to *integration* (*Figure 4B and C*, left). The model also predicts shorter decision times because it requires high SNR; the probability of detecting an extremum on each sample is exceedingly high when the stimulus is strong, such that the decision is made within a few samples. Given shorter decision times, *extrema detection* must predict longer non-decision times in order to produce RT distributions that are similar to those produced by *integration* (*Figure 4B and C*, middle). Importantly, the difference in non-decision time is robust and typically ranges from 50 to 150 ms. Therefore, empirical constraints on the non-decision time should, in theory, disentangle the models.

Our results thus far show that many observations commonly taken as evidence for an integration strategy can also be explained by non-integration strategies. However, we also identified the factors that allow non-integration models to mimic *integration*. In the next section, we leverage these insights to design a task that disentangles the models and test whether subjects used an integration strategy to perform the task.

## A motion discrimination task that disentangles the models

The modeling exercises described above illustrate that *integration* and *extrema detection* differ in their predictions for SNR (determined by the $\kappa$ parameter) and non-decision time (in FR task-designs). This suggests that constraining the estimates of these parameters should cause the models to make predictions that are substantially different and hence testable. How can this be achieved experimentally? It is generally not possible to measure the SNR of the momentary evidence used by a subject, and an estimation of SNR from neural data relies upon assumptions about how sensory information is encoded and decoded (e.g. *Ditterich, 2006*). Instead, we reasoned that SNR in a FR task ought to be closely matched to that in a VSD task, so long as properties of the sensory stimulus are unchanged. Therefore, if a subject performs trials in both a VSD and a FR task-design, a model that accurately estimates SNR should be able to parsimoniously explain data from both trial-types with a single, common $\kappa$ parameter. A model that does not accurately estimate SNR would require two separate $\kappa$ parameters to explain data from both trial-types. To be even more stringent, a successful model should be able to fit data from one trial-type and use the estimated $\kappa$ parameter to accurately predict data from the other trial-type. We also reasoned that the non-decision time can be constrained and/or empirically estimated through conditions that minimize the decision time. If a

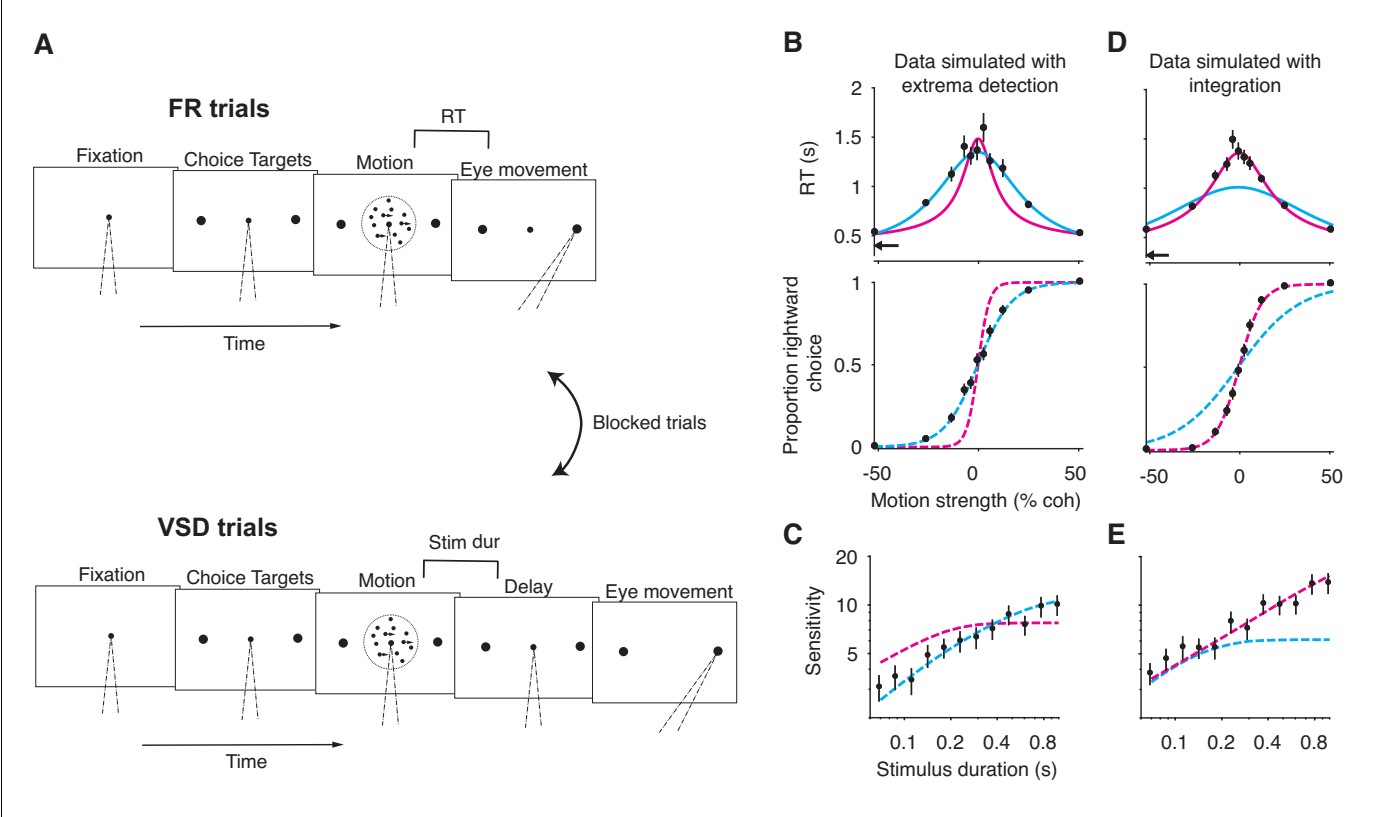

**Figure 5.** A motion discrimination task that disentangles the integration and extrema detection models. (**A**) Schematic of the task. The task requires subjects to judge the direction (left versus right) of a random-dot-motion (RDM) movie in blocks of free response (FR) and variable stimulus duration (VSD) trials. In a second task (not shown), subjects were presented with 100% coherent motion only and were instructed to respond as fast as possible while maintaining perfect accuracy. (**B**) Simulation of FR trials in the RDM task (N = 2145 trials, 165 trials per stimulus strength). Reaction time (top) and the proportion of positive choices (bottom) are plotted as a function of stimulus strength. Data (black points) were generated from an extrema detection model (same parameters as in *Figure 4*). Positive (negative) motion coherence corresponds to rightward (leftward) motion. The arrow shows the non-decision time used to generate the simulated data. Solid curves are model fits to the mean RTs. Both models were constrained to use the data-generating non-decision time. Dashed curves are predictions using the parameters obtained from the RT fits. With the non-decision time constrained, only the data-generating model succeeds in fitting and predicting the data. (**C**) Simulation of VSD trials (N = 3000 trials). Data (black points) were generated with an extrema detection model, using the same κ parameter as in **B**. Dashed curves are model fits. Each model's κ parameter was fixed to the value estimated from fits to the FR data. (**D, E**) Same as in (**B**) and (**C**) with *integration* as the data-generating model.

subject makes decisions that are so automatic that the decision times are negligible, then the resulting RTs would reflect only the non-decision time and hence confer an empirical estimate of the non-decision time distribution. Additionally, because decision time generally decreases as the stimulus strength increases, the non-decision time can be constrained by including sufficiently strong stimulus strengths.

We incorporated these constraints into a random-dot-motion (RDM) discrimination task. The task requires subjects to judge the direction of motion in the RDM stimulus and report their decision by making an eye movement to a corresponding choice target (*Figure 5A*). We included blocks of trials that switched between a FR design and a VSD design and forced the models to fit data from the VSD trials using the κ parameter derived from fits to FR trials. We constrained the non-decision time in two ways: first, we interleaved a small proportion of 100% coherence trials, which contain completely unambiguous motion and thus require minimal decision time to render a correct choice. Second, we conducted a supplementary experiment in which subjects received trials that included only 100% coherent motion and were instructed to respond as quickly as possible while maintaining perfect accuracy. We refer to these trials as speeded decision trials. In these trials, decision time is minimized as much as possible, thereby giving rise to an empirical estimate of the non-

decision time mean and standard deviation. We forced the models to adopt these non-decision time estimates when fitting data.

Before describing the experimental results, we first verify, through simulations, that this task-design disentangles the models. *Figure 5B-E* shows model fits to simulated data generated by *extrema detection* and *integration* performing the task described above. The models can be clearly distinguished. First, only the data-generating model successfully predicted choice-data from mean RTs (*Figure 5B, D*). Interestingly, each model failed in a systematic way when it was not the data-generating model: the integration model produced an overly narrow RT function and overestimated the slope of the choice function; the extrema detection model produced an overly broad RT function and underestimated the slope of the choice function. Second, only the data-generating model successfully predicted sensitivity in the VSD trials when forced to use the $\kappa$ parameter derived from fits to the FR data (*Figure 5C and E*). Finally, a model comparison heavily favored the data-generating model when comparing fits to the full RT distributions ($\Delta \text{BIC} = -1172.0$ when *extrema detection* generated the data; $\Delta \text{BIC} = 1080.1$ when *integration* generated the data). We will use these analyses as benchmarks when we analyze data from the human subjects.

## The decision strategies of human subjects performing a motion discrimination task

Six human subjects performed the motion discrimination task. As expected, stronger motion led to more accurate and faster choices in FR trials (*Figure 6A*). In VSD trials, longer stimulus durations were associated with greater sensitivity (*Figure 6B*). The speeded decision trials gave rise to similar non-decision time estimates across the six subjects (black arrows, top row of *Figure 6A*; *Table 1*), and it succeeded in minimizing decision time, as RTs in this experiment were substantially faster than those for the 100% coherence trials that were interleaved in the main experiment (*Figure 6—figure supplement 1*). As predicted by our results in the first part of the paper, if we did not include the constraints on the non-decision time and SNR, *integration* and *extrema detection* could not be clearly differentiated on the basis of their fits to—and predictions of—the subjects' data (*Figure 6—figure supplement 2*).

With constraints on the non-decision time and $\kappa$ parameter, *extrema detection* was incompatible with each subject's dataset. *Figure 6A* shows the mean RT fits (solid curves) and the corresponding choice-data predictions (dashed curves) for both models. We also fit a logistic function to the choice-data alone (black curves), which approximates an upper-limit on the quality of the choice-accuracy predictions given binomial variability and the assumption that choices are explained by a logistic function of motion coherence. *Extrema detection* produced visibly poor fits to the mean RTs in many cases, and in all cases fit the mean RTs worse than *integration* (*Table 1*). Additionally, the model systematically underestimated the slope of the choice function when predicting choice-data. In other words, the subjects' choice-accuracy was too high to be explained by an extrema detection model, given their pattern of decision times.

We next asked whether the extrema detection model can accurately fit the VSD data using the $\kappa$ parameter estimated from fits to the FR data. While the variants of the models we used to fit the mean RTs are parsimonious in that they use only four free parameters, they may not yield the most accurate estimates of the $\kappa$ parameter. Therefore, we estimated $\kappa$ with an elaborated version of each model in which the decision-bounds can symmetrically collapse toward zero as a function of time. The collapsing decision-bounds allow the models to fit the full RT distributions while accounting for some features in the data that are not accounted for by the parsimonious model (e.g. longer RTs on error trials; see *Ditterich, 2006*). These estimates of $\kappa$ are shown in *Table 2* (top row), and were generally similar to the values estimated by the parsimonious model (*Table 1*, top row). With $\kappa$ constrained, the extrema detection model's fits to the VSD trials were visibly poor for four of the six subjects (dashed cyan curves in *Figure 6B*). In the remaining two subjects (S3 and S5), *extrema detection* produced a reasonable fit to the VSD data. Nevertheless, for every subject, *extrema detection* failed at least one of the benchmarks described above.

In contrast, we found strong evidence in favor of the integration model for some of the subjects. For subjects 1-3, the model's ability to fit and predict data despite rigid constraints was remarkable. First, the predicted choice function closely resembled the fitted logistic function (*Figure 6A*), and the log-likelihood of the prediction was statistically indistinguishable from that of the logistic fit in two of these subjects (S1: $p = 0.012$; S2: $p = 0.052$; S3: $p = 0.058$, bootstrap). Second, *integration*

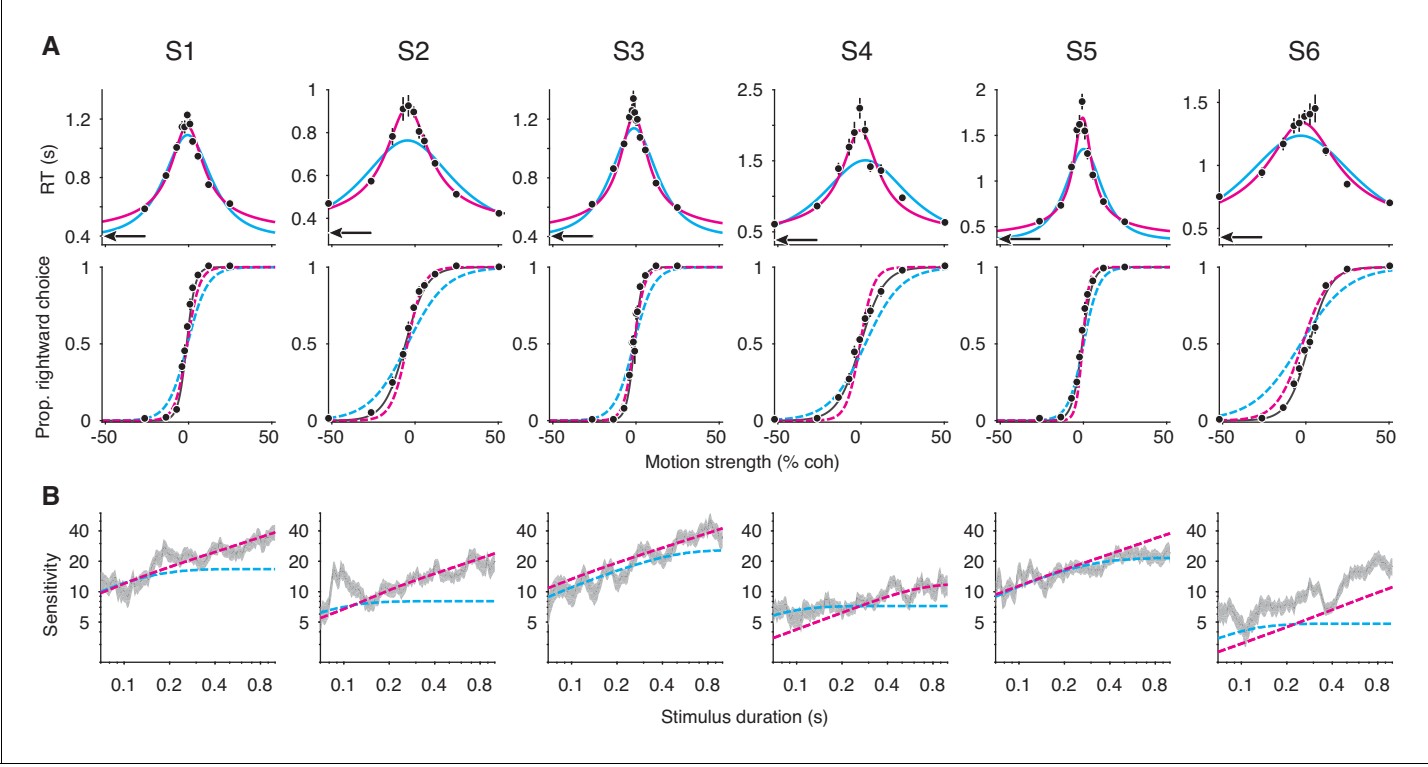

**Figure 6.** Failure of the extrema detection model and mixed success of the integration model in human subjects performing the RDM task. (**A**) Free response (FR) trials. Reaction times (top) and proportion of rightward choices (bottom) as a function of motion strength for six human subjects (error bars are s.e.; ~1800 trials per subject). Positive (negative) motion coherence corresponds to rightward (leftward) motion. The arrows (top) indicate each subject's estimated non-decision time from the speeded decision trials. Solid curves are model fits to the mean RTs (*integration*, magenta; *extrema detection*, cyan). Dashed curves are model predictions using the parameters obtained from the RT fits. Solid black lines (bottom) are fits of a logistic function to the choice-data alone. For visualization purposes, data from the interleaved 100% coherence trials are not shown (see *Figure 6—figure supplement 1*). (**B**) Variable stimulus duration (VSD) trials (~3000 trials per subject). Sensitivity as a function of stimulus duration for the same subjects as in (**A**). Sensitivity is estimated using logistic regression (sliding logarithmic time window, see Materials and methods). Dashed curves are model fits to the VSD data, using the $\kappa$ parameter derived from model fits to the FR data.

The online version of this article includes the following source data and figure supplement(s) for figure 6:

**Source data 1.** Data from the RDM task for each subject, separated by trial-type.

**Figure supplement 1.** The reaction times from the 'speeded decision' experiment provide a better estimate of the non-decision time than those from the interleaved, 100% coherence trials.

**Figure supplement 2.** Without constraints on the non-decision time and SNR, the models cannot be easily distinguished.

could accurately fit data from VSD trials using the $\kappa$ parameter derived from fits to the FR data (*Figure 6B*). Finally, *integration* was heavily favored over *extrema detection* when fitting the full RT distributions (*Table 2*). We thus find clear support for an integration strategy in three of the human subjects.

For subjects 4-6, the evidence in favor of the integration model was less compelling. The model overestimated the slopes of the choice functions for subjects 4 and 5 (dashed curves, bottom row of *Figure 6A*), and these predictions were worse than those of the extrema detection model (*Table 1*). In subject 6, integration offered a reasonable prediction for the slope of the psychometric function, but inaccurately predicted the subject's choice-bias. The fit of the integration model to these subjects' VSD data also produced mixed results (*Figure 6B*). The integration model could fit VSD data from subject 5 using the $\kappa$ parameter derived from the fits to the FR data, but, as mentioned above, so could the extrema detection model. And, in subjects 4 and 6, constraining the $\kappa$ parameter caused integration to underestimate sensitivity as a function of stimulus duration. Despite these shortcomings, *integration* was heavily favored over *extrema detection* for these subjects based on fits to the full RT distributions (*Table 2*).

**Table 1.** Parameters of the integration and extrema detection models (with flat bounds) fit to mean RT data.
$\Delta$BICs are relative to the fit of the integration model. Positive values indicate that *integration* produced a better fit/prediction.

| Model | Integration | | | | | | Extrema detection | | | | | |
|---|---|---|---|---|---|---|---|---|---|---|---|---|
| Subject | S1 | S2 | S3 | S4 | S5 | S6 | S1 | S2 | S3 | S4 | S5 | S6 |
| $\kappa$ | 15.7 | 13.57 | 17.53 | 9.9 | 21.96 | 6.69 | 103.3 | 55.07 | 104.87 | 55.18 | 130.3 | 42.64 |
| B | 0.87 | 0.77 | 0.909 | 1.25 | 1.16 | 0.958 | 0.0757 | 0.0727 | 0.076 | 0.0786 | 0.078 | 0.0766 |
| $C_0$ | −0.008 | −0.0453 | −0.012 | −0.006 | −0.005 | −0.02 | −0.006 | −0.0448 | −0.0121 | 0.026 | 0.001 | −0.031 |
| $t_{ND}$ mean (empirical) | 0.39 | 0.326 | 0.394 | 0.367 | 0.351 | 0.42 | 0.39 | 0.326 | 0.394 | 0.367 | 0.351 | 0.42 |
| $\Delta$BIC: RT fit | 0 | 0 | 0 | 0 | 0 | 0 | 471.36 | 204.4 | 202.88 | 135 | 494.7 | 71.74 |
| $\Delta$BIC: choice prediction | 0 | 0 | 0 | 0 | 0 | 0 | 127.42 | 90.64 | 146.38 | −31.78 | −11.62 | 121.38 |

Thus far, we have primarily drawn conclusions based on how well the data conform to predictions made by *integration* and *extrema detection*, but this approach has its drawbacks. First, it implies a potentially false dichotomy between the two models. Second, the approach requires us to arbitrarily determine whether a model's predictions are good enough, because no reasonable model will ever perfectly predict all of the idiosyncrasies associated with real data. Finally, it is unclear what to conclude when neither model makes accurate predictions. Our results are a case in point: for subjects 1–3, the *integration* predictions were good but not always perfect; and, in subjects 4–6, the predictions of both models were mediocre. This invites a more nuanced approach.

*Integration* and *extrema detection* can be thought of as two ends of a continuum of sequential sampling models that differ in their degree of *leaky integration*. *Integration* posits a time-constant of infinity (i.e. no leak or perfect integration) and *extrema detection* posits a time-constant of zero (i.e. infinite leak). In a leaky integration model, the time-constant is a free parameter that lies between zero and infinity (***Busemeyer and Townsend, 1993***; ***Usher and McClelland, 2001***). The time-constant determines the rate at which the decision variable decays to zero with a lack of sensory input—that is, it determines how much information is lost as new information is acquired. It thus bestows the model with some flexibility over the relationship between decision-time and choice-accuracy. Our results give rise to two hypotheses in the context of the leaky integration model: (1) The model should support negligible information loss for the subjects who were well-explained by perfect integration. (2) The model should support non-negligible information loss for the subjects who could not be explained by either perfect *integration* or *extrema detection*.

**Table 2.** Parameters of the integration and extrema detection models with collapsing decision-bounds (see Materials and methods) fit to full RT distributions.
$\Delta$BIC values are relative to the fit of the integration model. Positive values indicate that *integration* produced a better fit.

| Model | Integration | | | | | | Extrema detection | | | | | |
|---|---|---|---|---|---|---|---|---|---|---|---|---|
| Subject | S1 | S2 | S3 | S4 | S5 | S6 | S1 | S2 | S3 | S4 | S5 | S6 |
| $\kappa$ | 19.79 | 11.86 | 21.91 | 7.97 | 19.11 | 5.88 | 120.2 | 60.84 | 170.91 | 55.38 | 146.52 | 36.26 |
| $B_0$ | 1.0828 | 0.698 | 1.159 | 1.15 | 1.08 | 0.98 | 0.0765 | 0.0739 | 0.085 | 0.0796 | 0.0791 | 0.0772 |
| a | 0.7 | 5.0 | 0.53 | 14.3 | 5.24 | 1.97 | 47.34 | 1.2 | 0.165 | 0.285 | 0.55 | 9.18 |
| d | 0.4 | 8.0 | 0.33 | 12.4 | 12.36 | 31 | 2.3 | 4.0 | 0.0005 | 49 | 49 | 32 |
| $\mu_{tnd}$ (empirical) | 0.39 | 0.326 | 0.394 | 0.367 | 0.351 | 0.42 | 0.39 | 0.326 | 0.394 | 0.367 | 0.351 | 0.42 |
| $\sigma_{tnd}$ (empirical) | 0.05 | 0.043 | 0.036 | 0.079 | 0.07 | 0.068 | 0.05 | 0.043 | 0.036 | 0.079 | 0.07 | 0.069 |
| $C_0$ | −0.0103 | −0.05 | −0.0099 | −0.002 | −0.0054 | 0.0139 | −0.0126 | −0.0746 | −0.0107 | −0.0035 | −0.0064 | 0.017 |
| $\Delta$BIC | 0 | 0 | 0 | 0 | 0 | 0 | 1,495.1 | 607.1 | 801.2 | 464.0 | 701.2 | 889.1 |

To obtain our best estimate of the integration time-constant, we fit the leaky integration model to the FR and VSD data simultaneously, thereby forcing the model to fit both datasets with a common $\kappa$ and leak parameter, and the decision-bounds were allowed to collapse toward zero over time. As before, the model was also forced to adopt the empirical estimates of the non-decision time mean and standard deviation. All other parameters were allowed to take on any value and were allowed to vary across the two trial-types. With this fitting protocol, we could faithfully recover the parameters that generated simulated datasets (*Figure 7—figure supplement 1*).

*Figure 7A* shows the model's estimated time-constant for each subject. For subjects 2 and 3, the time-constants were effectively infinite. As such, the leaky integration model was functionally equivalent to the perfect integration model. Consistent with this conclusion, a model comparison supported that the addition of a leak parameter was not justified for these subjects ($\Delta BIC > -10$; *Figure 7B*). The estimated time-constant for subject 1 was shorter than the time-constants for subjects 2 and 3. However, the ΔBIC indicates that the leak parameter was not strongly justified. Note the large Bayesian credible intervals for the estimated time-constants (*Figure 7A*, thick orange lines for the interquartile range, thin orange lines for the 95% credible interval). This is because the time-constant becomes less identifiable as its value approaches and exceeds the longest decision times. The estimated lower bounds for the time-constants are close to the decision times at the most difficult stimulus conditions, again suggesting that these subjects made decisions by integrating motion information with little to no information loss.

We found evidence for *leaky integration* in two of the remaining three subjects. In subjects 4 and 5, the model produced time-constants that were just below 1 s (0.80 s for subject 4, 0.84 s for

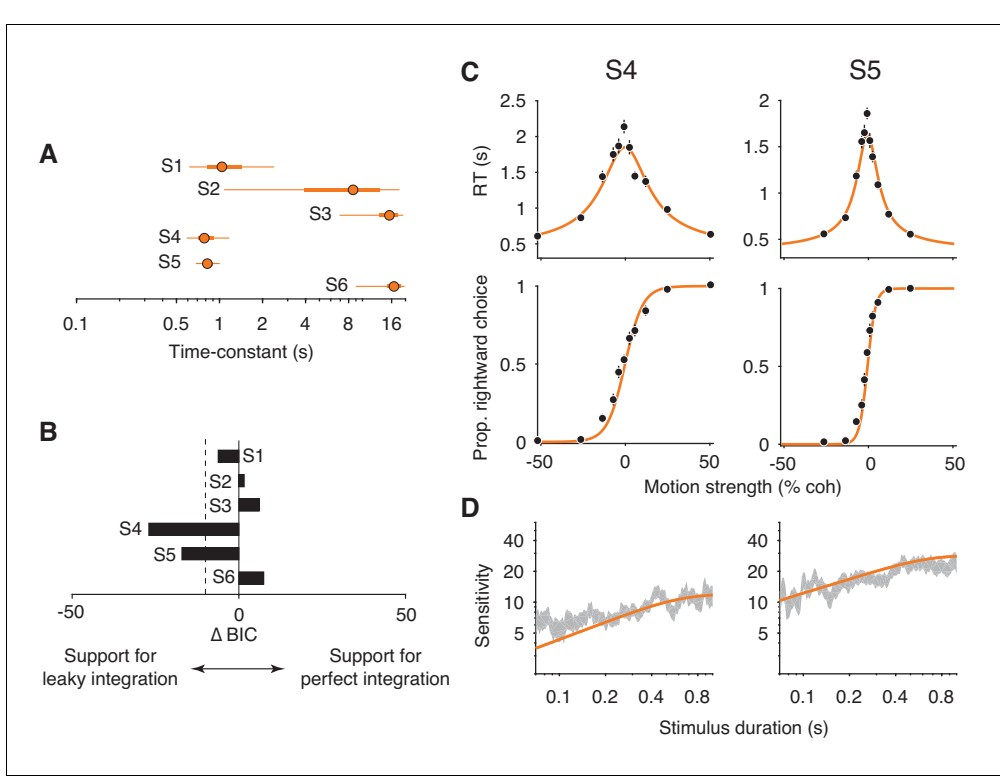

**Figure 7.** Fits of a leaky integration model. (**A**) Integration time-constants estimated with a leaky integration model for each subject (S1–S6). The fits to FR and VSD data were constrained to share a common κ parameter. Thick and thin lines represent 50% and 95% Bayesian credible intervals, respectively, estimated by Variational Bayesian Monte Carlo (VBMC; *Acerbi, 2018*). (**B**) Comparison of the leaky integration and perfect integration models. Negative ΔBIC values indicate support for the leaky integration model. The leaky integration model is supported in S4 and S5 (ΔBIC < −10; dashed line). (**C**) Leaky integration model fits for FR trials from S4 and S5. Data are the same as in *Figure 6*. (**D**) Leaky integration model fits for VSD trials from the same subjects as in (**C**). The online version of this article includes the following figure supplement(s) for figure 7:

**Figure supplement 1.** Parameter recovery for the leaky integration model.

subject 5; *Figure 7A*) and the addition of the leak parameter substantially improved the quality of the model fit (*Figure 7B*). *Figure 7C and D* (orange curves) shows these fits, which capture the main features of both datasets. Note, however, that the model still slightly underestimated the sensitivity of subject 4 for the shortest stimulus durations. Finally, the leaky integration model failed to account for data from subject 6. The fitted time-constant was indistinguishable from infinity (*Figure 7A*) and thus the failure of the perfect integration model (*Figure 6*, subject 6) could not be accounted for by leak.

## Discussion

We considered three general classes of decision-making strategies—sequential sampling with integration (e.g. drift diffusion), sequential sampling without integration (e.g. extrema detection), and no sequential sampling and no integration (e.g. snapshot)—and found that disentangling integration and non-integration strategies is more difficult than previously appreciated. Simulations of these models in different task-designs showed that several observations interpreted as conclusive evidence for integration were also predicted by non-integration strategies. Additionally, the integration model consistently fit simulated data well, even when these data were generated by non-integration models. Together, these results demonstrate the ease with which behavioral data could be misattributed to an integration mechanism.

We are not the first to propose that non-integration mechanisms can explain behavior in perceptual decision-making tasks. In fact, the first model that attempted to reconcile accuracy with reaction time resembles our extrema detection model (*Cartwright and Festinger, 1943*). Similar non-integration mechanisms, known as *probability summation*, have long been used to explain the detection of static stimuli as a function of their intensity, duration, and spatial properties (e.g. *Sachs et al., 1971*; *Watson, 1979*; *Robson and Graham, 1981*). A parallel line of research showed that bounded evidence integration is optimal for decisions based on sequences of independent samples (*Barnard, 1946*; *Wald, 1947*; *Good, 1979*). Such integration models also reconcile accuracy with reaction time (e.g. *Stone, 1960*; *Ratcliff, 1978*). Given these insights, one might naturally assume that subjects integrate evidence when making decisions about stochastic stimuli, which comprise independent samples of evidence. This assumption can be problematic, however, if untested. For as we show here, integration and non-integration models can behave similarly.

In the first part of the paper, we identified several factors that allow non-integration models to mimic integration. A crucial factor was the freedom of the models to fit the SNR of the momentary evidence to the data. Non-integration models are highly suboptimal compared to integration and therefore require higher SNR to produce the same level of performance. They are free to adopt this relatively large SNR because the true SNR cannot be measured directly. In other words, there is a trade-off between the SNR of the momentary evidence and the efficiency with which evidence samples are combined. Integration and non-integration models account for the same data by assuming different balances between the two. Of course, only one balance between SNR and efficiency holds in reality for a given dataset, and this is why the models can be disentangled if the SNR is adequately constrained. We demonstrate that the SNR can be adequately constrained if its estimate is derived from a separate task-design (see also *Drugowitsch et al., 2016*).

In FSD and VSD task-designs, we found that non-integration models mimicked *integration* in part because of a guessing mechanism. If the stimulus extinguished before an extremum was detected or a sample was acquired, then the decision was based on a coin-flip. This guessing rule allowed *extrema detection* to produce a range of psychophysical kernel shapes in a FSD task and to improve sensitivity with increasing stimulus duration in a VSD task. An alternative to guessing is to base the choice on the final sample, and we show that this variant of the model does not mimic *integration*. Therefore, it is possible to rule-out this variant based on data that conform to *integration*, but the same may not hold for *extrema detection* with a guessing rule.

In agreement with *Ditterich, 2006*, we found that *extrema detection* mimics *integration* in a FR task in part because the non-decision time is unconstrained. Given fits to the same dataset, *extrema detection* predicts longer non-decision time than *integration*. This observation is a consequence of its exponentially distributed decision times and the aforementioned requirement of higher SNR, both of which require the non-decision time to compensate for the relatively short decision times. The difference in predicted non-decision time between the two models manifests at strong stimulus

strengths—that is, conditions that minimize decision time. If these conditions are excluded from experimental designs, then the models can evade punishment for inaccurate estimates of the non-decision time.

The results from the first part of the paper illustrate the difficulty of ruling out non-integration strategies. There are several implications. At the very least, experimenters should not assume subjects integrate evidence just because integration is the optimal strategy. The results also imply that a given behavioral observation should be adduced as conclusive evidence for an integration strategy only if it is not reproduced by alternative strategies. Notably, non-integration models mimicked *integration* for reasons that were often counterintuitive, which stresses the importance of testing the predictions of alternative models through simulations (see also *Palminteri et al., 2017*; *Wilson and Collins, 2019*). Similarly, our findings discourage experimenters from drawing strong conclusions about decision strategy or an underlying neural mechanism based on the quality of a model fit, without first verifying that the model fails to fit data from conceptually opposing mechanisms. The practices that our results caution against are relatively standard in the field. Indeed, our own group has used these practices in previous work to support claims about subjects' decision strategies. It would be prudent to consider potential alternative strategies when designing experiments in order to ensure that behavioral data is not misattributed to an integration mechanism.

Such misattribution could lead to a variety of errors when neural data is used to make inferences about the neural mechanisms of integration. For example, if data generated by a non-integration strategy were misattributed to an integration strategy, an experimenter might mistake short bursts of neural activity for a mechanism of integration, or they might conclude that a brain area does not integrate because its activity does not reflect an integration process. In cases where neural activity is perturbed, brain areas that are essential to evidence integration might not be identified as causal. This is not to say that neural activity cannot, in its own right, inform models of the decision process. *Ditterich (2006)* used this approach to show that neural responses in the lateral intraparietal area are most consistent with an integration process that includes a time-variant gain. More broadly, if neural responses clearly reflect integrated evidence (e.g. *Huk and Shadlen, 2005*), then it would be reasonable to presume that the subjects' decisions were based on integrated evidence.

The misidentification of decision strategy could also be problematic when model fits to behavioral data are used to make inferences about underlying mechanisms. This approach is widely applied in mathematical psychology (*Ratcliff and McKoon, 2008*) and computational psychiatry (*Montague et al., 2012*). We showed that the misidentification of a subject's decision strategy leads to parameter fits that are systematically misleading. For example, fits of the integration model to simulated data generated by *extrema detection* led to an underestimate of both SNR and non-decision time. Therefore, it is possible that differences in best-fitting model parameters between experimental groups (e.g. patients vs. controls) do not actually reflect a difference in these parameters but in the strategies deployed by the two groups. A more explicit consideration of alternative strategies will add to our ability to link specific model components to underlying mechanisms. Indeed, it might reveal differences between experimental groups or conditions that would have otherwise gone undetected.

We wish to emphasize that integration models are often useful even in the absence of evidence for an integration strategy. They can be used as descriptive models of behavioral data. For example, they allow experimenters to estimate a subject's sensitivity while controlling for RT or characterize a subject's speed-accuracy trade-off. The model's use in these ways is similar to the use of signal detection theory models to derive a criterion-free estimate of sensitivity (*Green and Swets, 1966*). Furthermore, many studies on decision-making are ambivalent about whether subjects integrated evidence but can still use integration models to draw conclusions about other components of the decision process. For example, in *Kang et al. (2017)*, subjects performed a motion discrimination task and, after viewing the stimulus, adjusted the setting of a clock to indicate the moment they felt they had reached a decision. The authors used an integration model to conclude that aspects of these subjective decision times accurately reflected the time at which a decision-bound was reached. They fit the model to the subjective decision times and the resulting parameters could accurately predict subjects' choices. Another germane example is work from *Evans et al. (2018)*, who used an integration framework to study the neural mechanism of the decision bound in mice making escape decisions. They identified a synaptic mechanism that produced an all-or-nothing response in the dorsal periaqueductal grey when it received a critical level of input from the superior

colliculus, which caused the mice to flee. While our results suggest that neither study produced definitive behavioral evidence for integration, substituting *extrema detection* for *integration* would not change the studies' main conclusions.

In the second part of the paper, we used the insights derived from our simulations to design a version of the RDM discrimination task that constrained the SNR and non-decision time. The constraints allowed us to rule-out a non-integration strategy for each subject we tested, which is consistent with the idea that SNR in visual cortex would have to be implausibly high in order for a non-integration strategy to be viable in a RDM task (*Ditterich, 2006*). We found strong evidence for effectively perfect integration in half of our subjects. In these subjects, the predictions of the perfect integration model were remarkably accurate, even with strong parameter constraints. Data from two of the remaining three subjects appeared to lie somewhere between the predictions of perfect integration and no integration, and fits of a leaky integration model were consistent with this observation. The time-constants estimated by the leaky integration model suggested that these subjects integrated with only minimal information loss. Surprisingly, no model we tested offered a satisfactory explanation of the data from subject 6. The failure of the models in this subject nonetheless reinforces the fact that the models were not guaranteed to succeed in the other subjects.

We accounted for some of the failures of the perfect integration model with a leaky integration model; however, we suspect some other models that do not posit leaky integration could do so as well. Examples include models that only integrate strong evidence samples (*Cain et al., 2013*), competing accumulator models with mutual inhibition but no leak (*Usher and McClelland, 2001*), and models that posit noise in the integration process itself (*Drugowitsch et al., 2016*). The shared feature of these models and leaky integration is that they involve information loss. We focused on leaky integration because, for those who seek to understand how information is maintained and manipulated over long time-scales, substantial leakage would be most problematic. With this in mind, the fact that the 'leaky' subjects yielded time-constants on the order of half a second to a second is encouraging. At the very least, they were integrating information over time-scales that are substantially longer than the time-constants of sensory neurons and the autocorrelation times of the visual stimulus. Furthermore, the time-constants are likely underestimated. Our estimates of the non-decision time are biased toward longer times because we assume decision time is negligible in the speeded decision experiment, and an overestimate of the non-decision time would lead to an underestimate of the time-constant.

While the small number of subjects we used prevents us from making sweeping claims, the apparent variability in decision strategy across our subjects underscores the importance of analyzing data at the level of individuals. Many of our findings would be obfuscated had we not analyzed each subject separately. Our insights are also relevant to an ongoing debate about whether subjects' decisions are better explained by an urgency-gating model (*Cisek et al., 2009*; *Thura et al., 2012*; *Carland et al., 2015a*; *Carland et al., 2015b*), which posits little to no integration, or a drift-diffusion model (*Winkel et al., 2014*; *Hawkins et al., 2015*; *Evans et al., 2017*). A subject's strategy could lie somewhere between no integration and perfect integration or in a completely different space of models. A subject may also change their strategy depending on several factors, including the task structure, the nature of the stimulus, and the subject's training history (*Brown and Heathcote, 2005* ; *Evans and Hawkins, 2019*; *Tsetsos et al., 2012*; *Glaze et al., 2015*; *Ossmy et al., 2013*). Further characterization of the factors that affect decision strategy will be an important direction for future work.

Of course, our approach is not the only one available to rule-out non-integration strategies. For example, *Pinto et al. (2018)* tasked mice with counting the number of visual 'pulses' that appeared as the mice ran through a virtual corridor. The authors showed that a snapshot strategy predicted a linear psychometric function under certain conditions in their task, which did not match the mice's behavioral data. Additionally, *Waskom and Kiani (2018)* were able to rule-out a non-integration strategy for humans performing a contrast discrimination task. Discrete evidence samples were drawn from one of two possible distributions, and subjects chose which generating distribution was most likely. Because the distributions overlapped, there was an upper bound on the performance of any strategy that utilized only a single sample, and subjects performed better than this upper bound. This approach could be used in similar task-designs (e.g. *Drugowitsch et al., 2016*). Finally, as mentioned above, *Ditterich (2006)* showed that the SNR in visual cortex would have to be implausibly high in order for a non-integration strategy to explain data from a RDM discrimination task. These

examples and others (e.g. *Glickman and Usher, 2019*) illustrate that the best approach for ruling-out non-integration strategies will likely depend on the specifics of the stimulus and the task-design.

Nevertheless, any attempt to differentiate integration from non-integration strategies requires that the latter be considered in the first place. Here, we demonstrated the importance of such consideration, identified why non-integration strategies can mimic integration, and developed an approach to rule them out. The general approach should be widely applicable to many evidence integration tasks, although it will likely require modifications. It explicitly mitigates the factors that allow non-integration strategies to mimic integration and allow integration models to fit data generated by alternative mechanisms. By doing so, non-integration strategies can be ruled-out, and the predictions of evidence integration models can be tested in a regime where they can reasonably fail. We hope that our insights help lead to more precise descriptions of the processes that underlie decisions and, by extension, cognitive processes that involve decisions. Such descriptions will enhance our understanding of these processes at the level of neural mechanism.

## Materials and methods

### Description of the models

We explored four decision-making models. A shared feature of all four models is that they render decisions from samples of noisy momentary evidence. We model the momentary evidence as random values drawn from a Normal distribution with mean $\mu = \kappa(C - C_0)$ and unit variance per second, where $\kappa$ is a constant, $C$ is the stimulus strength (e.g. coherence), and $C_0$ is a bias term. We implement bias as an offset to the momentary evidence because the method approximates the normative solution under conditions in which a range of stimulus strengths are interleaved, such that decision time confers information about stimulus strength (see *Hanks et al., 2011*). Note that each model receives its own set of parameter values. Each strategy differs in how it uses momentary evidence to form a decision.

#### Integration

We formalized an integration strategy with a drift-diffusion model. The model posits that samples of momentary evidence are perfectly integrated over time. The expectation of the momentary evidence distribution is termed the drift rate, and its standard deviation is termed the diffusion coefficient. The decision can be terminated in two ways: (i) The integrated evidence reaches an upper or lower bound ($\pm B$), whose sign determines the choice; (ii) The stream of evidence is extinguished, in which case the sign of the integrated evidence at that time determines the choice. Note that only (i) applies in a FR task-design because the stimulus never extinguishes before a decision-bound is reached.

To estimate the predicted proportion of positive (rightward) choices as a function of stimulus strength and duration in a VSD task, we used Chang and Cooper's finite difference method (1970) to numerically solve the Fokker-Planck equation associated with the drift-diffusion process. We derive the probability density of the integrated evidence ($x$) as a function of time ($t$), using a $\Delta t$ of 0.5 ms. We assume that $x = 0$ at $t = 0$ (i.e., the probability density function is given by a delta function, $p(x,\ t = 0) = \delta(0)$). At each time step, we remove (and keep track of) the probability density that is absorbed at either bound. The proportion of positive (rightward) choices for each stimulus duration is therefore equal to the density at $x>0$ at the corresponding time point. We fit the model parameters ($\kappa,\ B, C_0$) to VSD data by maximizing the likelihood of observing each choice given the model parameters, the signed stimulus strength, and the stimulus duration. Unless otherwise stated, we used Bayesian adaptive direct search (BADS; *Acerbi and Ji, 2017*) to optimize the model parameters. All model fits were confirmed using multiple sets of starting parameters for the optimization. Unless stated otherwise, when fitting VSD data from the 'constrained' RDM task (see below; e.g. *Figure 6*), instead of fitting the $\kappa$ parameter, we used the $\kappa$ parameter estimated from fits of an elaborated, collapsing bound model to FR data (see below).

For the FR task, we used two variants of the model. The first is more parsimonious, and assumes that the decision bounds are flat (as above). This variant of the model provides analytical equations for the predicted proportion of positive choices and mean RT:

$$P_+ = \left[ 1 + e^{-2\kappa(C - C_0)B} \right]^{-1} \tag{1}$$

$$t_{\mathrm{R}} = \frac{B}{\kappa(C - C_0)} \tanh[\kappa(C - C_0)B] + t_{\mathrm{ND}} \tag{2}$$

where $t_{ND}$ is the mean non-decision time, which summarizes sensory and motor delays. *Equation 2* allowed us to fit the model's parameters to the mean RTs, which we then used to predict the psychometric function with *Equation 1*. Note that *Equation 2* explains the mean RT only when the sign of the choice matches the sign of the drift rate (for nonzero stimulus strengths). To account for this, we identified trials to be included in the calculation of mean RTs by first finding the point of subjective equality (PSE), given by a logistic function fit to choices. We then only included trials whose choice matched the sign of the stimulus strength, adjusted by the PSE. The PSE was not taken into account when fitting the $C_0$ parameter. The parameters in *Equation 2* ($\kappa, B, C_0, t_{ND}$) were fit by maximizing the log-likelihood of the mean RTs given the model parameters, assuming Gaussian noise with standard deviation equal to the standard error of the mean RTs. Optimization was performed using MATLAB's *fmincon* function.

We also used a more elaborate variant of the model that allowed the decision-bound to collapse toward zero over time in order to explain full, choice-conditioned RT distributions. In principle, the elaborated model should provide a more precise estimate of $\kappa$ because it takes into account all the data instead of just the mean RTs. The model also explains features of the data that are not explained by the flat-bounds model (e.g. longer RTs on error trials). In the elaborated model, the bounds remained symmetric around zero but were a logistic function of time:

$$B(t) = \pm B_0 \left( 1 + e^{a(t-d)} \right)^{-1} \tag{3}$$

where $a$ is constrained to be nonnegative. The predicted decision time distribution for each stimulus strength and choice was derived by computing the probability density of the integrated evidence that exceeded each decision bound. We convolved these decision time distributions with a Gaussian non-decision time distribution with mean $\mu_{\mathrm{tnd}}$ and standard deviation $\sigma_{\mathrm{tnd}}$ in order to generate the predicted RT distributions. The model parameters ($\kappa, B_0, a, d, C_0, \mu_{\mathrm{tnd}}, \sigma_{\mathrm{tnd}}$) were fit by maximizing the log-likelihood of the observed RT on each trial given the model parameters, the stimulus strength, and the choice.

## Extrema detection

In the extrema detection model, each independent sample of momentary evidence is compared to a positive and negative detection threshold or bound ($\pm B$). As with *integration*, the decision can be terminated in two ways: (i) A sample of momentary evidence exceeds $\pm B$, in which case the decision is terminated and the choice is determined by the sign of $\pm B$. (ii) The stream of evidence is extinguished. In the latter case, we implemented two different rules for determining the choice. The main rule we implemented posits that the choice is randomly determined with equal probability for each option. We believe this 'guess' rule is most appropriate because the essence of the model is that evidence is ignored if it does not exceed a detection threshold. We also explored a rule in which the choice is determined by the sign of the last sample of evidence acquired before the stimulus extinguished, which we termed a 'last sample' rule (*Figure 3—figure supplement 1B*). Note that *extrema detection* and *integration* had the same set of free parameters and the procedures for fitting the two models were identical.

The behavior of the model is primarily governed by the probability of exceeding $\pm B$ on each sample. These probabilities are described by,

$$p_{+B} = \frac{1}{2} \mathrm{erfc}\left( \frac{B - \mu \Delta t}{\sqrt{2\Delta t}} \right), \tag{4}$$

$$p_{-B} = \frac{1}{2} \mathrm{erfc}\left( \frac{B + \mu \Delta t}{\sqrt{2\Delta t}} \right) \tag{5}$$

where erfc is the complementary error function. The equations represent the density of the momentary evidence distribution that exists beyond $\pm B$. We assumed $\Delta t = 0.5$ ms and adopted the same variance per time-step as in the integration model. The probability of a positive choice, conditional on an extremum being detected, is therefore,

$$P(+\text{choice}|E) = \frac{p_{+B}}{p_{+B} + p_{-B}}, \tag{6}$$

where $E$ signifies an extremum was detected. Note that we use uppercase $P$ to represent probabilities associated with trial outcomes and lowercase $p$ to represent probabilities associated with single samples of evidence. The probability of a positive choice, conditional on the stimulus extinguishing before an extremum is detected, depends on the choice rule. For the guess rule this probability is 0.5. For the last sample rule it is the probability that the sample was greater than zero and did not exceed $\pm B$:

$$P_{\text{guess}}(+\text{choice}|\neg E) = \frac{1}{2}, \tag{7}$$

$$P_{\text{last\_sample}}(+\text{choice}|\neg E) = \frac{\int_0^B M dx}{\int_{-B}^B M dx}. \tag{8}$$

where $M$ is the momentary evidence distribution. Note that none of the equations above depend on the passage of time (i.e., the number of samples). However, the cumulative probability of detecting an extremum does increase with the number of samples and is described by a cumulative geometric distribution:

$$P_E = 1 - (1 - p_{\pm B})^N \tag{9}$$

where $N$ is the number of samples ($N = \lceil t/\Delta t \rceil$). The fact that *Equation 9* depends on the number of samples, as opposed to time, is potentially important. It means that the cumulative probability of detecting an extremum depends not only on $p_{\pm B}$, but also $\Delta t$. We used a $\Delta t = 0.5$ ms in order to match that used in the integration model, and the results were unchanged when we used $\Delta t = 1$ ms. However, the behavior of the model could change with changes to the sampling rate. Combining *Equation 6* through 9 gives us the probability of a positive choice as a function of stimulus duration in a VSD experiment:

$$P(+\text{choice}|t) = P_E P(+\text{choice}|E) + (1 - P_E) P(+\text{choice}|\neg E). \tag{10}$$

In a FR experiment, the decision can only be terminated if an extremum is detected. Therefore, the predicted proportion of positive choices is given by *Equation 6*. The predicted mean RT, in seconds, is described by,

$$t_R = \Delta t \frac{1}{p_{\pm B}} + t_{ND}. \tag{11}$$

Similar to the procedure for fitting the integration model, Equation 11 was used to fit mean RT data and the resulting model parameters were plugged into *Equation 6* to generate a predicted choice function.

As with the integration model, we used an elaborated model with collapsing decision bounds to explain the full, choice-conditioned RT distributions. We used *Equation 3* to parameterize the collapsing bounds. The collapsing bounds cause the probability of detecting an extremum on each sample to increase with time. The probability of the decision terminating after $N$ samples (i.e. the decision time distribution), regardless of choice, is described by,

$$P_N = p_{\pm B(N)} \prod_{n=1}^{N-1} \left(1 - p_{\pm B(n)}\right) \tag{12}$$

where $p_{\pm B(n)}$ is the probability of exceeding a decision-bound on sample $n$. Intuitively, the cumulative product represents the probability that an extremum is not detected after $N-1$ samples. Note that

*Equation 12* simplifies to the geometric distribution when the decision-bounds are flat. The decision time density function conditional on choice is the product of the unconditioned decision time density function and the probability that a decision at each time-point will result in a positive (or negative) choice. These density functions are then convolved with a truncated Gaussian non-decision time probability density function to produce the predicted RT distributions (the truncation ensures that non-decision times are always positive).

## Snapshot

In the snapshot model, only a single sample of momentary evidence is acquired on each trial. We do not consider mechanisms that would determine when the sample is acquired. We instead assume that the time at which the sample is acquired is a predetermined random variable and is independent of the stimulus strength. The distribution that describes this random variable can be chosen arbitrarily. For simplicity, the sampling times were assumed to be uniformly distributed in the FSD task. We used an exponential distribution for the sampling times in the VSD task, although several other distributions produced similar results. If the sample is acquired before the stimulus extinguishes, then the choice is determined by the sample's sign. Otherwise, the choice is randomly assigned with equal probability for each option. The probability of a positive choice when a sample is acquired is described by,

$$P(+\text{choice}|S) = \frac{1}{2}\text{erfc}\left(\frac{\mu}{\sqrt{2}}\right). \tag{13}$$

The overall probability of a positive choice as a function of viewing duration is then,

$$P(+\text{choice}|t) = P_S P(+\text{choice}|S) + \frac{1}{2}(1 - P_S) \tag{14}$$

where $P_S$ is the probability of acquiring a sample. It is a function of viewing duration and depends on the distribution of sampling times. While *Equation 14* resembles what is described for *extrema detection* (*Equation 10*), there is a crucial difference: unlike *extrema detection*, the probability that a choice is based on evidence is independent of the stimulus strength. In a FR task, the probability of a positive choice is governed by *Equation 13*. The predicted RT distribution is simply the distribution of sampling times convolved with the non-decision time distribution, and it is independent of the stimulus strength.

## Leaky integration

The leaky integration model is a simple extension of the (perfect) integration model. The model posits that the rate of change of the decision variable depends on both the momentary evidence and its current value, the latter of which causes it to decay exponentially or 'leak' toward zero if input is withdrawn. The decay's half-life is determined by a single parameter, which is termed the integration time-constant, $\tau$. The shorter the time-constant the more the decision variable 'leaks.' Perfect *integration* and *extrema detection* can be thought of as special cases of the leaky integration model, in which the time-constant is infinite and zero, respectively. The decision variable, $x$, is modeled as an Ornstein-Uhlenbeck (O-U) process, such that,

$$dx = (-\lambda x + \mu)dt + \xi\sqrt{dt} \tag{15}$$

where $\lambda = \tau^{-1}$ and $\xi$ is the standard Wiener process.

We developed a method to derive the probability density function of the integrated evidence for the leaky integration model. As for the perfect integration model, we assume that $x = 0$ for $t = 0$ (i.e. probability density function of the integrated evidence is given by a delta function, $p(x, t = 0) = \delta(0)$). First, we propagate the probability density function of the integrated evidence, $p(x, t)$, for one small time step ($\Delta t = 0.5$ ms). We use Chang and Cooper's implicit integration method (*Chang and Cooper, 1970*; *Kiani and Shadlen, 2009*), assuming perfect integration from $t$ to $t + \Delta t$ (i.e., $\lambda = 0$). We then add the influence of the leak, through a linear transformation that maps the probability of the integrated evidence being $x$ at time $t + \Delta t$, to a new value of integrated evidence, $x'$, where $x' = xe^{-\Delta t/\lambda}$. This shrinks the probability density function toward zero in proportion to the

leak parameter, $\lambda$. We iterate the two-step process until the motion stimulus is turned off or until the probability mass that has not been absorbed at either bound becomes negligibly small. As described for the *integration* model, at each time step we remove (and keep track of) the probability mass that is absorbed at either bound.

We estimated the posterior of the model parameters in order to determine the range of time-constants that best explain each subject's data (*Figure 7A*). Numerically calculating the posterior is computationally expensive. Instead, we calculated approximate posterior distributions of our model parameters with Variational Bayesian Monte Carlo (VBMC; *Acerbi, 2018*), which uses variational inference and active-sampling Bayesian quadrature to approximate the posterior distributions. We used highly conservative priors over the parameters when estimating the posteriors and changes to the priors had negligible effects within a large range.

We also performed a parameter recovery analysis to verify that our fitting procedure and the VBMC method accurately estimated ground-truth parameters used to generate data from the leaky integration model. We simulated the constrained RDM task to produce nine datasets, each of which were generated with a unique combination of $\kappa$, $B$, and $\tau$ (see *Table 3*). Each simulation contained ~3000 FR trials and 3000 VSD trials. We found that the approximate posteriors of the parameters, obtained through VBMC, accurately reflect these parameters (*Figure 7—figure supplement 1*).

## Model simulations

We simulated the *integration*, *extrema detection*, and *snapshot* models to compare the predictions they made in different task-designs. Each trial had a randomly chosen stimulus strength that remained constant throughout the trial's duration. We used stimulus strengths that mimicked those commonly used in a RDM discrimination task: ±.512, ±.256, ±.128, ±.064, ±.032, and 0. This allowed us to calibrate our model parameters such that the simulated data resembled real data from RDM tasks.

In the simulations of FSD experiments, each trial contained a transient stimulus pulse, which we used to calculate a psychophysical kernel for each dataset. The pulse added or subtracted 0.1 units of stimulus strength for 100 ms, thereby shifting the mean of the momentary evidence distribution for that duration. After the 100 ms pulse, the stimulus strength returned to its original value. The sign of the pulse was random and the timing of its onset was uniformly distributed in steps of 100 ms starting at $t = 0$. The psychophysical kernel is described by the relationship between the time of the pulse and its effect on choice across all trials, which we estimated with a logistic regression such that

$$P_+ = \left[1 + e^{-X\beta}\right]^{-1}, \tag{16}$$

**Table 3.** Generating parameters used for the parameter recovery analysis (*Figure 7—figure supplement 1*).

Each row represents the set of three parameters used to simulate data with the leaky integration model.

| Simulation | Model parameters | | |
| --- | --- | --- | --- |
| | $\kappa$ | $B$ | $\tau$ |
| 1 | 15 | 0.5 | 0.1 |
| 2 | 14.63 | 0.55 | 0.15 |
| 3 | 14.25 | 0.59 | 0.23 |
| 4 | 13.88 | 0.65 | 0.36 |
| 5 | 13.5 | 0.71 | 0.55 |
| 6 | 13.13 | 0.77 | 0.84 |
| 7 | 12.75 | 0.84 | 1.28 |
| 8 | 12.38 | 0.92 | 1.96 |
| 9 | 12 | 1 | 3 |

where $X$ is a design matrix. The design matrix included a column for each pulse-onset time, which took the form of a signed indicator variable ($X_{pulse} \in \{-1, 0, 1\}$). We also included a column for the trial's stimulus strength, although results were similar if this was not included. Each row of the design matrix therefore summarizes the stimulus strength and the pulse-onset time for a given trial. The ordinate in *Figure 2B* is the value of $\beta$ associated with each pulse-onset time.

The simulations of VSD experiments were identical to the FSD simulations, except there were no pulses and the stimulus duration on each trial was randomly chosen from a list of 12 possible, logarithmically-spaced durations between 0.07 s and 1.0 s. Each simulation therefore yielded 12 psychometric functions—one for each stimulus duration. To calculate a measure of sensitivity for each stimulus duration, we fit each psychometric function with a logistic function and used the fitted slope parameter to summarize sensitivity.

In the FR simulations, the stimulus remained on until a decision bound was reached. We did not simulate the snapshot model in a FR task. For the model comparisons in *Figure 4—figure supplement 2*, we generated 600 FR datasets, half of which were generated by *integration* and the other half of which were generated by *extrema detection*. We varied the number of simulated trials among 10, 100, and 1000 trials per stimulus condition, such that there were 100 datasets per model per trial count. For each model, we pseudorandomly chose 100 sets of generating parameters within a range of plausible parameter values (Integration: $5<\kappa<25, 0.6<1.2, 0.3<\mu_{\mathrm{tnd}}<0.4, 0.02<\sigma_{\mathrm{tnd}}<0.08$; Extrema detection: $50<\kappa<215, 0.07<0.08, 0.46<\mu_{\mathrm{tnd}}<0.56, 0.09<\sigma_{\mathrm{tnd}}<0.11$). The same 100 sets of generating parameters were used across all three trial groups.

For the graphs in *Figure 5*, we simulated each model in a FR design and a VSD design using the same $\kappa$ parameter for the two designs. The number of simulated trials in each design was similar to that collected for the human subjects (~2000 total FR trials; ~3000 total VSD trials). The FR simulation also included stimulus strengths of ±0.99 with the stimulus strengths listed above.

## Random dot motion task

We explored the decision strategies of human subjects with a 'constrained' random-dot-motion (RDM) discrimination task. The subjects were required to make a binary choice about the direction of motion of randomly moving dots. The RDM movies were generated using methods described previously (*Roitman and Shadlen, 2002*). Three interleaved sets of dots were presented on successive video frames (75 Hz refresh rate). Each dot was redrawn three video frames later at a random location within the stimulus aperture or at a location consistent with the direction of motion; the motion coherence is the probability of the latter occurring, and it remained constant throughout the duration of the trial. Note that even though the coherence does not fluctuate within a trial, the effective motion strength (e.g. motion energy) at each time point does fluctuate due to the stochastic nature of the stimulus (see *Zylberberg et al., 2016*). The stimulus aperture subtended 5° of visual angle, the dot density was 16.7 dots/deg²/s, and the size of the coherent dot-displacement was consistent with apparent motion of 5 deg/s. Stimuli were presented on a CRT monitor with the Psychophysics toolbox (*Brainard, 1997*). Subjects' eye positions were monitored with a video tracking system (Eyelink 1000; SR Research, Ottawa, Canada).

Six subjects (five male and one female) performed the task. One subject (S3) is an author on this paper. Another subject (S1) had previous experience with RDM stimuli but was naive to the purpose of the experiment. The remaining four subjects were naive to the purpose of the experiment and did not have previous experience with RDM stimuli. Each of these four subjects received at least one training session (~1000 trials) before beginning the main experiment to achieve familiarity with the task and to ensure adequate and stable task performance.

The main experiment consisted of two trial-types, VSD and FR, which were presented in blocks of 150 and 100 trials, respectively. Each subject performed ~4800 trials in total across 4-6 sessions, yielding ~3000 VSD trials (S1: 2946 trials; S2: 3007; S3: 2656; S4: 3086; S5: 3095; S6: 3069) and ~1800 FR trials (S1: 1833 trials; S2: 1866; S3: 1814; S4: 1766; S5: 1831; S6: 1914). Subjects initiated a trial by fixating on a centrally located fixation point (0.33° diameter), the color of which indicated the trial-type (red for VSD trials, blue for FR trials). Two choice targets then appeared on the horizontal meridian at 9° eccentricity, one corresponding to leftward motion and one corresponding to rightward motion. After 0.1 to 1 s (sampled from a truncated exponential distribution with $\tau = 0.3$ sec), the RDM stimulus appeared, centered over the fixation point. In VSD trials, subjects were required to maintain fixation throughout the stimulus presentation epoch. Once the stimulus extinguished,

subjects reported their choice via an eye movement to the corresponding choice-target. Fixation breaks before this point resulted in an aborted trial. In order to ensure that subjects could not predict the time of stimulus offset, the stimulus duration on each trial was randomly drawn from a truncated exponential distribution (0.07-1.3 s, $\tau = 0.4$ sec). To account for the fact that the first three video frames contain effectively 0% coherent motion (see above), we subtracted 40 ms from the stimulus durations when modeling the VSD data (*Figure 6B*; *Figure 7D*). Doing so generally led to better model predictions; our conclusions are unchanged if we do not subtract the 40 ms. We assume that this 40 ms duration is accounted for by the non-decision time in FR trials. In FR trials, subjects were free to indicate their choice at any point after stimulus onset and RT was defined as the time spanning the stimulus onset and the indication of the choice. Additionally, ~7% of FR trials contained 100% coherent motion. Subjects received auditory feedback about their decision on every trial, regardless of trial-type, and errors resulted in a timeout of 1 s. Choices on 0% coherence trials were assigned as correct with probability 0.5.

At the end of their final session, subjects also performed a block of 300 to 400 FR trials, comprising only 100% coherent motion. Subjects were instructed to respond as fast as possible while maintaining perfect performance. This supplemental experiment was designed to reduce decision times as much as possible. If decision times were negligible, the resulting RTs would approximate each subject's non-decision time distribution. We used the mean and standard deviation of this distribution as the non-decision time parameters when fitting the models to data from the main experiment (see above). In practice, the decisions presumably take a very short, but non-negligible, amount of time. Thus, this 'empirical' non-decision time distribution probably overestimates the mean of the non-decision time, albeit slightly. Note that an overestimate of the non-decision time would induce an underestimate of the integration time-constant. As such, its use is conservative with respect to a claim that a subject is integrating over prolonged timescales.

## Statistical analysis

We quantified the quality of a model fit using the Bayesian information criterion (BIC), which takes into account the complexity of the model. The BIC is defined as

$$\mathrm{BIC} = k \cdot \ln(n) - 2\hat{L} \tag{17}$$

where $n$ is the number of observations, $k$ is the number of free parameters, and $\hat{L}$ is the log-likelihood of the data given the best-fitting model parameters. To compare the fits of two models, we report the difference of the BICs. Note that because *integration* and *extrema detection* have the same number of parameters, their $\Delta$BIC is equivalent to the difference of the deviance of the models. We treated 'pure' model predictions (e.g., predicting the choice-data from mean RT fits), as model fits with zero free parameters.

To evaluate the slope of a psychometric function and a reasonable upper-limit on the quality of the model-predicted psychometric functions, we fit the choice-data with a logistic function, in which the proportion of rightward choices is given by

$$P_{\mathrm{right}} = \left[1 + e^{-(\beta_0 + \beta_1 C)}\right]^{-1} \tag{18}$$

where $\beta_0$ determines the left-right bias and $\beta_1$ determines the slope of the psychometric function. This function represents an upper limit under the assumption that choices are governed by a logistic function of coherence and binomial noise. To test whether the model prediction is significantly worse than this upper limit, we used a bootstrap analysis. For each subject, we generated 10,000 bootstrapped choice-datasets and fit each bootstrapped dataset with the logistic function above. We then compared the resulting distribution of log-likelihood values with the log-likelihood of the model prediction. The quality of the prediction was deemed significantly worse from that of the logistic fit if at least 95% of the bootstrapped log-likelihoods were greater than the log-likelihood produced by the model prediction.

## Acknowledgements

This research was supported by the Howard Hughes Medical Institute, the National Eye Institute (R01 EY011378, EY013933), the National Institute of Neurological Disorders and Stroke (NS113113), and the Israel Institute for Advanced Studies. We thank the members of the Shadlen lab for helpful discussions.

## Additional information

### Funding

| Funder | Grant reference number | Author |
|---|---|---|
| Howard Hughes Medical Institute | | Ariel Zylberberg<br>Michael N Shadlen |
| National Eye Institute | EY011378 | Gabriel M Stine<br>Ariel Zylberberg<br>Michael N Shadlen |
| National Eye Institute | EY013933 | Gabriel M Stine |
| National Institute of Neurological Disorders and Stroke | NS113113 | Gabriel M Stine<br>Ariel Zylberberg<br>Michael N Shadlen |
| Israel Institute for Advanced Studies | | Michael N Shadlen |

The funders had no role in study design, data collection and interpretation, or the decision to submit the work for publication.

### Author contributions

Gabriel M Stine, Conceptualization, Resources, Software, Formal analysis, Validation, Investigation, Visualization, Methodology, Writing - original draft, Writing - review and editing; Ariel Zylberberg, Conceptualization, Resources, Software, Formal analysis, Supervision, Methodology, Writing - review and editing; Jochen Ditterich, Conceptualization, Supervision, Writing - review and editing; Michael N Shadlen, Conceptualization, Resources, Formal analysis, Supervision, Funding acquisition, Writing - original draft, Writing - review and editing

### Author ORCIDs

Gabriel M Stine (ID) https://orcid.org/0000-0003-4906-0461
Ariel Zylberberg (ID) http://orcid.org/0000-0002-2572-4748
Michael N Shadlen (ID) http://orcid.org/0000-0002-2002-2210

### Ethics

Human subjects: The institutional review board of Columbia University (protocol #IRB-AAAL0658) approved the experimental protocol, and subjects gave written informed consent.

### Decision letter and Author response

Decision letter https://doi.org/10.7554/eLife.55365.sa1
Author response https://doi.org/10.7554/eLife.55365.sa2

## Additional files

### Supplementary files

• Transparent reporting form

### Data availability

The data generated during this study are included in the source data file for Figure 6.

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
