## [Decision Letter]

**Acceptance summary:**

Your work addresses an important issue with the standard methodology for modeling perceptual decisions. Your careful simulations of non-integration models show very clearly that non-integration strategies can produce fits that are qualitatively similar to evidence integration (and therefore mimic evidence integration) in widely used paradigms. The novel methodology you propose to distinguish integration from non-integration strategies represents a timely achievement. Altogether, your work provides an important cautionary tale for the existing literature on modeling perceptual decisions. We congratulate you again for this work, which we are happy to publish in *eLife*.

**Decision letter after peer review:**

Thank you for submitting your article "Differentiating between integration and non-integration strategies in perceptual decision making" for consideration by *eLife*. Your article has been reviewed by two peer reviewers, including Valentin Wyart as the Reviewing Editor and Reviewer #1, and the evaluation has been overseen by Michael Frank as the Senior Editor. The following individual involved in review of your submission has agreed to reveal their identity: Marius Usher (Reviewer #2).

The reviewers have discussed the reviews with one another and the Reviewing Editor has drafted this decision to help you prepare a revised submission.

This manuscript describes a behavioral modeling study which aims at differentiating integration from non-integration strategies during perceptual decision-making. For this purpose, the authors rely first on a simulation-based approach by comparing an evidence integration model to two non-integration models (extrema detection and snapshot) in different paradigms used to study perceptual decision-making: fixed stimulus duration (FSD), variable stimulus duration (VSD) and free response (FR). The authors show that non-integration models display qualitative features commonly used as signatures of evidence integration. Based on the results of these simulations, the authors then propose a paradigm combining FR and VSD trials which afford to better distinguish integration from non-integration strategies. The authors report that six human subjects tested in their paradigm are better fitted (both quantitatively and qualitatively) by an evidence integration model.

Both reviewers found that your article addresses an important issue with the standard methodology for modeling perceptual decisions. Indeed, because evidence integration is an optimal (or at least adequate) cognitive strategy for such tasks, it is typically assumed that perceptual decisions rely on evidence integration. Your simulations of non-integration models clearly show that non-integration strategies (e.g., extrema detection) can produce behavior and fits that are qualitatively similar to evidence integration. Furthermore, the paradigm and methodology you propose to distinguish integration from non-integration strategies represents a timely achievement. Both reviewers found your article to be clearly written, and to provide an important cautionary tale for the existing literature on modeling perceptual decision-making. The fact that it emphasizes the importance of simulating competing models of behavior and performing parameter recovery analyses (as proposed by Palminteri et al., 2017 and Wilson and Collins, 2019) is also something very valuable for the field.

Although the reviewers do not have significant reservations that would require essential revisions of your article, they have identified different points (listed below) that would benefit from clarifications in a revised version of your article.

1) Origin of the behavioral similarity of integration and non-integration strategies:

The critical reason why non-integration strategies can mimic integration when fitted to behavioral data in random-dot motion paradigms could be brought up more explicitly in the manuscript. It appears to be that the motion evidence SNR is not measurable directly, and thus the SNR parameter in non-integration models can be set to widely implausible values to fit the behavioral data. The authors rightfully mention at the end of the Discussion that paradigms such as the ones used by Waskom and Kiani, 2019, but also Drugowitsch et al., 2016, afford to measure the sensory SNR and thus put an upper bound on the performance of any non-integration model. The issue of the non-measurability of the motion evidence SNR in random-dot motion paradigms could be stated earlier and more explicitly in the manuscript. It would make even clearer why non-integration models can be tweaked to fit behavioral data simulated using evidence integration.

2) Distinction between differences sources of behavioral variability:

Another related point of discussion could be the definition of SNR in the model. Noise in perceptual decision-making can arise from at least three different sources – as laid out in Drugowitsch et al., 2016: 1. noise in sensory processing (here, motion processing), 2. noise during evidence integration, and 3. noise during response selection. As emphasized in the previous point, the core issue put forward by the authors – that non-integration models can be fitted to data by tweaking a parameter (SNR) that is not measurable independently – illustrates the danger of not characterizing and quantifying the different sources of decision errors in these tasks. It could be useful to state explicitly in the Discussion that an alternative strategy for ruling out non-integration models is to measure the sensory SNR for the evidence used in the perceptual decision-making task. While it is not (at least not easily) possible for random-dot motion stimuli, it is clearly possible for pulsed evidence in terms of gratings (e.g., Drugowitsch et al., 2016), but also contrast (Waskom and Kiani, 2019).

3) Additional discussion of existing literature on integration and non-integration strategies:

i) The mechanism referred to as “extrema detection” may precede integration as an account of perceptual decisions, under the name of Probability Summation over Time (PST; Watson, 1979). While this is cited, it would be helpful to discuss in more detail why PST was historically used in psychophysical tasks, while evidence integration is preferred in studies with stochastic evidence that extends over longer intervals. Unless the authors believe that the two mechanisms were confounded in these older psychophysical studies and that, in fact, evidence integration was misidentified as evidence for PST.

ii) There is at least one recent study that provides a complementary method for distinguishing evidence integration and extrema detection (Glickman and Usher, 2019). The idea was to plot the "integrated evidence" until response in a FR task. As shown in Figure 4D of this paper, extrema detection predicts that the integrated evidence should increase with time. This is quite different for how the integrated evidence varies with time under integration mechanisms (Figure 4A, 4B), where it is either constant or decreasing. If the authors can record the time-varying motion evidence within each trial, they could compare their method to this one. Even without such comparison, a discussion of this complementary approach would be helpful.

4) Decision bounds:

There is some confusion due to the swap between fixed or collapsing bounds within the manuscript. The original comparison is presented using a fixed boundary model, but then suddenly the collapsing boundary is preferred. If the use of collapsing boundary is necessary for the demonstration, the article should mention this model in the first part. Also, one should provide more details about how it fares when compared to the fixed boundary model in terms of model evidence (e.g., BIC).

Another important point for existing research concerns how much model misspecification might impact parameter recovery with respect to decision bounds? Many applications of the drift-diffusion model are used for asking questions about how threshold is adjusted, e.g. to speed-accuracy manipulations, and how that might be altered in different groups (aging, ADHD, OCD, etc.). It would be very important to know, based on additional simulation-based analyses, whether the conclusions about decision thresholds/bounds are somewhat less impacted by model misspecification (i.e., whether there is true integration or not) than non-decision times.

5) Leaky integration:

The authors state that "leakiness" can be seen as a spectrum that links evidence integration with extrema detection. I agree that it is the case in FR paradigms which are modeled with an accumulation-to-bound model. But I can't see how it would be the case for bound-free evidence integration in FSD or VSD paradigms. Could the authors clarify this statement?

Also, the classification of S1 as non-leaky is not particularly convincing, especially since the difference in BIC seems to support leak for this particular subject (and its time constant is similar to that of S4 and S5).

Also, the authors should also mention somewhere in the Discussion that random-dot motion stimuli do not afford to distinguish between a time-dependent leak (as a function of time) and a stimulus-dependent leak (as a function of the presentation of evidence samples). Indeed, recent results obtained by Waskom and Kiani suggest that the "leakiness" observed during perceptual decision-making is stimulus-dependent, not time-dependent.

---

## [Author Response]

1) Origin of the behavioral similarity of integration and non-integration strategies:The critical reason why non-integration strategies can mimic integration when fitted to behavioral data in random-dot motion paradigms could be brought up more explicitly in the manuscript. It appears to be that the motion evidence SNR is not measurable directly, and thus the SNR parameter in non-integration models can be set to widely implausible values to fit the behavioral data. The authors rightfully mention at the end of the Discussion that paradigms such as the ones used by Waskom and Kiani, 2019, but also Drugowitsch et al., 2016, afford to measure the sensory SNR and thus put an upper bound on the performance of any non-integration model. The issue of the non-measurability of the motion evidence SNR in random-dot motion paradigms could be stated earlier and more explicitly in the manuscript. It would make even clearer why non-integration models can be tweaked to fit behavioral data simulated using evidence integration.

The reviewers are correct that the SNR, which cannot be measured directly, can be set to allow non-integration strategies to mimic integration. We stress, however, that this is not the only reason why non-integration models can mimic integration. High SNR is necessary for mimicry, but it is not the only factor required to reproduce most of the behavioral observations discussed in the paper.

The reviewers are also correct that the SNR needed to explain data from the RDM task with a non-integration model is likely implausible (as suggested by Ditterich, 2006). However, we would like to clarify that we can only make this judgement of implausibility because of the decades of work on how motion in RDM stimuli is represented in the primate visual cortex. There are many cases in which an experimenter cannot reasonably determine a plausible range for the SNR; for example, when using novel stimuli or less-studied animal models.

We agree that the points raised about SNR could be more explicit in the paper. To this end, we have made the following changes:

1) We added an inset to Figure 2A, which displays the SNR parameter for each model simulation. We hope this makes clear that the non-integration models require much higher SNR to produce the same choice behavior as integration.

2) We added the text, “In many cases, the SNR cannot be measured directly and thus the true SNR is generally unknown.”

3) When we introduce the motion discrimination task, we added the text, “It is generally not possible to measure the SNR of the momentary evidence used by a subject, and an estimation of SNR from neural data relies upon assumptions about how sensory information is encoded and decoded (e.g. Ditterich, 2006).”

4) In the Discussion, we added, “Non-integration models […] are free to adopt this relatively large SNR because the true SNR cannot be measured directly.”

We would like to clarify that the upper bound in performance that can be calculated in Waskom and Kiani, 2018 and Drugowitsch et al., 2016, is due to the fact that the distributions that generate the evidence samples overlap. It is an upper-bound because it is calculated assuming no sensory (i.e., internal) noise, and the upper-bound will depend on the degree to which the distributions overlap. We agree that this is an appealing approach to rule-out non-integration strategies. We clarified the approach in the Discussion:

“Waskom and Kiani, 2018, were able to rule-out a non-integration strategy for humans performing a contrast discrimination task. […] This approach could be used in similar task-designs (e.g., Drugowitsch et al., 2016).”

2) Distinction between differences sources of behavioral variability:Another related point of discussion could be the definition of SNR in the model. Noise in perceptual decision-making can arise from at least three different sources – as laid out in Drugowitsch et al., 2016: 1. noise in sensory processing (here, motion processing), 2. noise during evidence integration, and 3. noise during response selection. As emphasized in the previous point, the core issue put forward by the authors – that non-integration models can be fitted to data by tweaking a parameter (SNR) that is not measurable independently – illustrates the danger of not characterizing and quantifying the different sources of decision errors in these tasks. It could be useful to state explicitly in the Discussion that an alternative strategy for ruling out non-integration models is to measure the sensory SNR for the evidence used in the perceptual decision-making task. While it is not (at least not easily) possible for random-dot motion stimuli, it is clearly possible for pulsed evidence in terms of gratings (e.g., Drugowitsch et al., 2016), but also contrast (Waskom and Kiani, 2019).

Again, we do not fully agree that unconstrained SNR is the “core issue” put forward in our paper (see the response above).

In our paper, we focus only on noise in sensory processing (i.e., the SNR of the momentary evidence distribution). We agree in general that noise can affect processing at several different stages, but for our purposes, the alternative forms of noise are not as relevant. First, limiting our comparison to only sensory noise makes sense because noise in the integration process does not apply for non-integration strategies. Obviously, characterizing and quantifying the noise during the integration process would be circular, as it would require that we assume there is an integration process to begin with. Second, because our subjects did not make errors in the easiest conditions, we can rule-out noise in response selection.

It is nonetheless possible that noise in the integration process could explain why the perfect integration model failed for some of our subjects. We show that this failure can be explained by leaky integration (for some subjects), but we list other forms of information loss that might also explain the failure of the perfect integration model. In this paragraph, we have added noisy integration to the list of alternative models:

“We accounted for some of the failures of the perfect integration model with a leaky integration model; however, we suspect some other models that do not posit leaky integration could do so as well. […] The shared feature of these models and leaky integration is that they involve information loss.”

Concerning the point about SNR in RDM tasks and pulsed tasks, it is important to distinguish between the SNR of the stimulus (i.e., external noise) and the SNR of the momentary evidence distribution (i.e., the internal representation of the stimulus). The former can be measured directly and sometimes used to calculate an upper bound on performance with a non-integration strategy (as described in the previous response). Indeed, this analysis might be easier in paradigms where evidence is pulsed. The SNR of the momentary evidence distribution cannot be measured directly, even if evidence is pulsed. It must be estimated. We show that non-integration models can mimic integration when this estimate is poorly constrained. The SNR of the momentary evidence distribution can be adequately constrained if its estimate is derived from a separate task-paradigm. We demonstrate this for RDM stimuli. Drugowitsch et al., 2016, use a similar approach for “pulsed” grating stimuli. We have edited the relevant part of the paragraph to clarify these points. It now reads:

“Non-integration models […] are free to adopt this relatively large SNR because the true SNR cannot be measured directly. […] We demonstrate that the SNR can be adequately constrained if its estimate is derived from a separate task-paradigm (see also Drugowitsch et al., 2016).”

3) Additional discussion of existing literature on integration and non-integration strategies:i) The mechanism referred to as “extrema detection” may precede integration as an account of perceptual decisions, under the name of Probability Summation over Time (PST; Watson, 1979). While this is cited, it would be helpful to discuss in more detail why PST was historically used in psychophysical tasks, while evidence integration is preferred in studies with stochastic evidence that extends over longer intervals. Unless the authors believe that the two mechanisms were confounded in these older psychophysical studies and that, in fact, evidence integration was misidentified as evidence for PST.

Extrema detection is conceptually similar to PST, but there are important differences. Watson’s PST model was meant to explain a yes/no detection task and extrema detection is meant to explain discrimination tasks. Furthermore, PST assumes that it is effectively impossible to detect an extremum when the stimulus strength is equal to zero. Of course, we do not make this assumption in the extrema detection model, and this difference causes the behavior of the models to diverge drastically. Finally, extrema detection is most similar to the model proposed by Cartwright and Festinger, 1943, which precedes PST by 30 years.

We agree with the reviewers that extrema detection was heavily inspired by PST. We have amended the paragraph, in which we introduce the extrema detection model, to make this clearer. The beginning of the paragraph now reads:

“The first non-integration model we considered was extrema detection (Figure 1B). The model was inspired by probability summation over time (Watson, 1979), which was proposed as an explanation of yes-no decisions in a detection task. Extrema detection is also similar to other previously proposed models (Cartwright and Festinger, 1943; Ditterich, 2006; Cisek et al., 2009; Brunton et al., 2012; Glickman and Usher, 2019).”

We have no reason to believe that any specific older psychophysical study misattributed evidence integration to a non-integration process (e.g., PST), or vice versa. Our point is that one cannot directly assume that subjects integrate evidence over time just because the task demands it; this assumption has to be tested against non-integration strategies.

We agree that a discussion of the history of non-integration and integration models would be helpful. We have added the following paragraph to the Discussion:

“We are not the first to propose that non-integration mechanisms can explain behavior in perceptual decision-making tasks. […] For as we show here, integration and non-integration models can behave similarly.”

ii) There is at least one recent study that provides a complementary method for distinguishing evidence integration and extrema detection (Glickman and Usher, 2019; Cognition). The idea was to plot the "integrated evidence" until response in a FR task. As shown in Figure 4D of this paper, extrema detection predicts that the integrated evidence should increase with time. This is quite different for how the integrated evidence varies with time under integration mechanisms (Figure 4A, 4B), where it is either constant or decreasing. If the authors can record the time-varying motion evidence within each trial, they could compare their method to this one. Even without such comparison, a discussion of this complementary approach would be helpful.

We thank the reviewers for pointing us toward this paper. It is an interesting approach that we had not considered. To determine whether this approach would work for RDM stimuli, we simulated the stimulus with statistics that resembled those reported by Zylberberg et al., 2016, for the motion energy in RDM stimuli. We simulated 10,000 trials with both models. We ensured that the models produced similar choice and RT-functions and that these functions resembled those we observed in our data. We then calculated the integrated evidence at the time of the decision (with a sliding window) for both models, using only the external evidence signal and ignoring internal noise, as in Glickman and Usher, 2019. Author response image 1 shows the results of this analysis. We found that the analysis does not distinguish the models in our case. Both models predict that the integrated evidence should increase with decision time. The integration model predicts this because it is in a “high internal noise” regime (Glickman and Usher, 2019), which is the relevant regime for RDM tasks. We now refer to Glickman and Usher, 2019: “These examples and others (e.g., Glickman and Usher, 2019) illustrate that the best approach for ruling-out non-integration strategies will likely depend on the specifics of the stimulus and the task-design.”

4) Decision bounds:There is some confusion due to the swap between fixed or collapsing bounds within the manuscript. The original comparison is presented using a fixed boundary model, but then suddenly the collapsing boundary is preferred. If the use of collapsing boundary is necessary for the demonstration, the article should mention this model in the first part. Also, one should provide more details about how it fares when compared to the fixed boundary model in terms of model evidence (e.g., BIC).

We agree that the rationale behind switching to the collapsing bounds model could be clearer. Indeed, its introduction in the original manuscript was sudden and did not provide a rationale: “Note that, when estimating κ from fits to the [free response (FR)] data and comparing the quality of the fits, we used an elaborated variant of the models in which the decision-bounds can collapse symmetrically toward zero as a function of time (see below).” We have removed this sentence and have elected to introduce the collapsing bounds model in the next section. The first mention of the collapsing bounds model now reads:

“While the variants of the models we used to fit the mean RTs are parsimonious in that they use only four free parameters, they may not yield the most accurate estimates of the κ parameter. […] These estimates of κ are shown in Table 2 (top row), and were generally similar to the values estimated by the parsimonious model (Table 1, top row).”

We have also added text to the Materials and methods that explains the rationale for the collapsing bounds model:

“We also used a more elaborate variant of the model that allowed the decision-bound to collapse toward zero over time in order to explain full, choice-conditioned RT distributions. In principle, the elaborated model should provide a more precise estimate of κ because it takes into account all of the data instead of just the mean RTs. The model also explains features of the data that are not explained by the flat-bounds model (e.g., longer RTs on error trials). In the elaborated model, the bounds remained symmetric around zero but were a logistic function of time…”

The primary reason for the collapsing bounds model was to estimate κ for the VSD predictions. The fact that estimates of κ were similar across the collapsing bounds model and the flat bounds model suggests that the VSD predictions would also be similar across the two variants of the models. Furthermore, none of the results in the first part of the paper depend on collapsing bounds. We state early on that, “we assumed flat decision-bounds in the integration and extrema detection models, unless stated otherwise.” In the manuscript, the collapsing bounds constitute a minor technical issue – not a point of comparison between integration and non-integration models. The nature of the decision bound is of general interest to us and many others in the field, but a formal comparison of the collapsing bounds model and a flat bounds model is not germane to the paper and would distract readers from its message.

Another important point for existing research concerns how much model misspecification might impact parameter recovery with respect to decision bounds? Many applications of the drift-diffusion model are used for asking questions about how threshold is adjusted, e.g. to speed-accuracy manipulations, and how that might be altered in different groups (aging, ADHD, OCD, etc.). It would be very important to know, based on additional simulation-based analyses, whether the conclusions about decision thresholds/bounds are somewhat less impacted by model misspecification (i.e., whether there is true integration or not) than non-decision times.

We thank the reviewers for bringing up this point. Indeed, parameter fits of the drift-diffusion model are often used to interpret the effects of speed-accuracy manipulations. Both models change the speed-accuracy trade-off by changing the decision bounds. The effect of this change on the RT and choice functions is similar for both models. In theory, this means that an increase (or decrease) of the decision-bound across conditions should be identified as such, even with model misspecification. We make the more general point that integration models can be useful even in the absence of evidence for an integration strategy, so long as integration is not critical to the questions being asked, and we provide examples of such cases. We believe that within-subject analyses concerning changes in the speed-accuracy trade-off generally fit this description as well. We changed the beginning of the aforementioned paragraph so it now reads:

“We wish to emphasize that integration models are often useful even in the absence of evidence for an integration strategy. They can be used as descriptive models of behavioral data. For example, they allow experimenters to estimate a subject’s sensitivity while controlling for RT or characterize a subject’s speed-accuracy trade-off. The model's use in these ways is similar to the use of signal detection theory models to derive a criterion-free estimate of sensitivity (Green and Swets, 1966).”

Across-group comparisons are more complex. Typical approaches involve fitting the drift-diffusion model to data from both groups (e.g., patients and controls) and comparing some aspect of the model fits. As we explain in the paper, such comparison could be misleading if the two groups used different strategies to perform the task. We suspect that conclusions about decision bounds are not immune to this issue, but we also suspect that all of this will strongly depend on the nature of the comparison.

5) Leaky integration:The authors state that "leakiness" can be seen as a spectrum that links evidence integration with extrema detection. I agree that it is the case in FR paradigms which are modeled with an accumulation-to-bound model. But I can't see how it would be the case for bound-free evidence integration in FSD or VSD paradigms. Could the authors clarify this statement?

We agree that a spectrum of leakiness does not perfectly link the two models, particularly for FSD and VSD paradigms. We did not mean to imply a perfect link in the text. We have revised the text as follows: "Integration and extrema detection models can be thought of as two ends of a continuum of sequential sampling models that differ in their degree of leaky integration.”

Also, the classification of S1 as non-leaky is not particularly convincing, especially since the difference in BIC seems to support leak for this particular subject (and its time constant is similar to that of S4 and S5).

We classified this subject as “non-leaky” because the |∆BIC| was less than 10, which is a commonly used criterion for BIC comparisons. We have added a dashed line to Figure 7B to denote this criterion. In addition, the estimated time-constant rivaled the longest decision times for this subject. We nonetheless acknowledge that the classification is somewhat arbitrary. We have softened the conclusions about S1 in the text:

“Figure 7A shows the model’s estimated time-constant for each subject. For subjects 2 and 3, the time-constants were effectively infinite. […] However, the ∆BIC indicates that the leak parameter was not strongly justified.”

Also, the authors should also mention somewhere in the Discussion that random-dot motion stimuli do not afford to distinguish between a time-dependent leak (as a function of time) and a stimulus-dependent leak (as a function of the presentation of evidence samples). Indeed, recent results obtained by Waskom and Kiani suggest that the "leakiness" observed during perceptual decision-making is stimulus-dependent, not time-dependent.

We do not fully understand the distinction between “stimulus-dependent” and “time-dependent” leak. To our understanding, Waskom and Kiani, 2018, did not mention this distinction explicitly. We accounted for some of the subjects with a leaky integration model, and we agree with the reviewers that we cannot identify the cause of the leak. However, to reiterate our response in (2) and the text, we could likely substitute the leaky integration model with many other models that incorporate information loss. Given that this caveat is explicit in the paper, we believe a discussion of the caveat raised by the reviewers would be redundant.